



# NDACC FTIR trace gas measurements at the University of Toronto Atmospheric Observatory from 2002 to 2020

Shoma Yamanouchi[1], Stephanie Conway[1], Kimberly Strong[1], Orfeo Colebatch[1], Erik Lutsch[1], Sébastien Roche[1], Jeffrey Taylor[2], Cynthia H. Whaley[3] and Aldona Wiacek[4]

[1]Department of Physics, University of Toronto, Toronto, M5S 1A7, Canada
[2]Nova Scotia Community College, Halifax, B3K 2T3, Canada
[3]Environment and Climate Change Canada, Victoria, V8W 2Y2, Canada
[4]Saint Mary's University, Halifax, B3H 3C3, Canada

*Correspondence to*: S. Yamanouchi (syamanou@physics.utoronto.ca)

**Abstract.** Nineteen years of atmospheric composition measurements made at the University of Toronto Atmospheric Observatory (TAO, 43.66°N, 79.40°W, 174 m.a.s.l.) are presented. These are retrieved from Fourier Transform InfraRed (FTIR) solar absorption spectra recorded with an ABB Bomem DA8 spectrometer from May 2002 to December 2020. The retrievals have been optimized for fourteen species: $O_3$, $HCl$, $HF$, $HNO_3$, $CH_4$, $C_2H_6$, $CO$, $HCN$, $N_2O$, $C_2H_2$, $H_2CO$, $CH_3OH$, $HCOOH$ and $NH_3$ using the SFIT4 algorithm. The measurements have been archived in the Network for Detection of

Atmospheric Composition Change (NDACC) data repository in Hierarchical Data Format version 4 (HDF4) files following the Generic Earth Observation Metadata Standard (GEOMS) and are also publicly available on Borealis, the Canadian Dataverse Repository. In this paper, we describe the instrumentation, the retrieval strategy, the vertical sensitivity of the retrievals, the quality assurance process, and error analysis of the TAO FTIR measurements, and present the current version of the time series.

## 1 Introduction

The University of Toronto Atmospheric Observatory (TAO) was established in 2001 in downtown Toronto, Ontario, Canada (43.66°N, 79.40°W) at 174 m above sea level (a.s.l.). The site was established to provide long-term high-quality measurements of trace gases for short-term and long-term scientific studies, satellite validation, and model comparisons and assessments. Due to its urban location, measurements made at TAO are affected by the densely populated area of the city of Toronto and

its surroundings; Toronto is the fourth largest city in North America and the Greater Toronto Area is the centre of Canada's largest population, which was 6.2 million in the 2021 census (Statistics Canada, 2022). Measurements are also affected by the industrial centres of the northeastern United States (Whaley et al., 2015). TAO is thus ideally located for the detection of tropospheric pollutants, and is also well suited for monitoring mid-latitude stratospheric ozone, related species, and greenhouse gases (Wiacek, 2007; Yamanouchi et al., 2021b). In addition, the site is well positioned to detect mid-latitude polar intrusion

events as shown in Whaley et al. (2013) and transported wildfire plumes (Lutsch et al., 2016; Yamanouchi et al., 2020)



The primary instrument at TAO is the ABB Bomem DA8, a high-resolution Fourier Transform InfraRed (FTIR) spectrometer, which has been acquiring daily measurements of solar absorption spectra since May 2002, weather permitting. In March 2004, the TAO site became part of the Network for Detection of Atmospheric Composition and Change (NDACC, De Mazière et al., 2018), affiliated with its InfraRed Working Group (IRWG), filling the North American mid-latitude gap in the network of FTIR stations. As a station in the NDACC IRWG, the following trace gases are routinely retrieved from the TAO solar absorption spectra: $O_3$, HCl, HF, $HNO_3$, $CH_4$, $C_2H_6$, CO, HCN and $N_2O$. In addition, a number of other trace gases are retrieved as research products and are also archived including $C_2H_2$, $H_2CO$ (Vigouroux et al., 2018), $CH_3OH$, HCOOH (Franco et al., 2021), $NH_3$ (Dammers et al., 2016; Yamanouchi et al., 2021a), and OCS (Hannigan et al., 2022). Finally a small number of species are retrieved as research products but on an event basis and are not yet included in the public archive, including $CCl_4$, NO, and $NO_2$. Select species retrieved at TAO (currently $CH_4$, CO, $H_2CO$, and $O_3$) are also contributed to the Copernicus Atmospheric Monitoring Service (CAMS) Rapid Delivery (RD) system (https://atmosphere.copernicus.eu/). Rapid Delivery species are archived within one month of making measurements, and consolidated datasets are archived to NDACC annually. Descriptions of the instrument, retrievals, and studies using TAO measurements can be found in Wiacek (2006), Taylor (2008), Whaley (2014), and Yamanouchi (2021).

TAO measurements have been included in multiple publications. They have been used to study meso-thermospheric NO (Wiacek et al, 2006), the time evolution of HCl and HF total column abundances (Kohlhepp et al., 2012), the transport of biomass burning plumes (Griffin et al, 2013; Palmer et al., 2013, Lutsch et al., 2016, 2020; Yamanounchi et al., 2020), mid-latitude polar vortex intrusions (Whaley et al., 2013) and pollution events (Whaley et al., 2015), emissions associated with oil and natural gas extraction (Franco et al., 2016; Tzompsa-Sosa et al., 2017, 2019), increases in atmospheric methane (Bader et al., 2017),and the impact of COVID-19-related emissions reductions on free tropospheric ozone (Steinbrecht et al., 2020) and on CO, $CO_2$, and $CH_4$ (You et al., 2021). Trends of all species retrieved at TAO (from 2002 to 2019) were presented in Yamanouchi et al. (2021a). Furthermore, TAO FTIR data have been used for instrument inter-comparisons (Wunch et al., 2007; Taylor et al., 2008; Sun et al., 2020; Yamanouchi et al., 2021b), and evaluation of chemical reanalysis of CO in a coupled chemistry-climate model (Gaubert et al., 2016). In addition, TAO measurements have been used in the validation of data products from satellite missions, including ACE (Mahieu et al, 2005; Clerbaux et al., 2008; De Maziere et al., 2008; Kerzenmacher et al., 2008; Mahieu et al., 2008; Strong et al., 2008; Dupuy et al., 2009), SCIAMACHY (Dils et al, 2006), OSIRIS (Taylor et al., 2007), IASI (Dammers et al., 2016; Franco et al., 2020), CrIS (Dammers et al., 2017), MOPITT (Buchholz et al., 2017), GOSAT (Olsen et al., 2017), and TROPOMI (Vigouroux et al., 2020; Sha et al., 2021).

This paper is organized as follows. In Section 2, the TAO facility, and in particular the FTIR instrument used to measure the time series presented in this work, is described. In Section 3, the data analysis procedure, error characterization, and quality control method used to process the solar absorption spectra are detailed. Finally in Section 4, the time series of $O_3$, HCl, HF, $HNO_3$ (four stratospheric NDACC species), $CH_4$, $C_2H_6$, CO, HCN, $N_2O$ (five tropospheric NDACC species), $C_2H_2$, $H_2CO$, $CH_3OH$, HCOOH and $NH_3$ (five research products) retrieved at TAO over the last 19 years are presented.



## 2 Instrumentation

### 2.1 TAO FTIR spectrometer

The primary instrument at the University of the Toronto Atmospheric Observatory is the TAO DA8, a high-resolution research-grade Fourier transform infrared spectrometer that measures solar absorption spectra in the mid-infrared (Wiacek et al., 2007). The instrument is installed in a laboratory on the 16th floor of the Burton Tower in the Department of Physics. The TAO FTIR spectrometer is coupled to a dedicated heliostat on the rooftop above the laboratory, which directs the solar beam through an opening to the instrument. A description of the solar tracker follows in Section 2.2.

The DA8 FTIR spectrometer was manufactured by ABB Bomem Inc. and features an improved mirror design to increase the stability of the optical alignment. The arrangement includes a dynamically aligned mirror mounted below the interferometer's moving mirror to maintain alignment of the optics, mitigating any off-axis motion of the moving mirror. This optical design fixes the optical axis through the beamsplitter and sets in place the focal point at the detector as shown in Figure 1.

The FTIR spectrometer has a maximum optical path difference of 250 cm, corresponding to a maximum apodized resolution of 0.004 cm$^{-1}$. It is equipped with a KBr beamsplitter and two detectors (InSb and HgCdTe (MCT)), which together provide spectral coverage from 700 to 4300 cm$^{-1}$. This region is routinely measured using the six standard NDACC IRWG optical interference bandpass filters mounted on a filter wheel. The characteristics of these filters, including their spectral coverage and the trace gases commonly measured with each, the substrate, the wedge angle, and manufacturer, are detailed in Table 1. Measurements are made sequentially through these filters over the course of the day. A recent study by Blumenstock et al. (2021) has quantified the channeling features associated with optical components used by NDACC FTIR instruments and found that the TAO system has a channeling fringe with a frequency of about 3 cm$^{-1}$ due to the InSb detector window, but is less affected than other sites by channeling associated with construction of the KBr beamsplitter. TAO FTIR solar absorption spectra are recorded throughout the day in a semi-autonomous manner with operator involvement for start-up, shutdown, and monitoring tasks. Due to the dependence on clear-sky conditions, FTIR measurements are typically limited to 80 to 100 days per year with some bias towards the summer months. Since February 28, 2014 the scan speed of the FTIR spectrometer has been increased from 0.5 cm/s to 1 cm/s, which has resulted in a larger number of measurements per day. Testing was performed prior to this change and no discernible impact on measured spectra or retrievals was found when the scan speed was increased.

The optical alignment of the TAO FTIR spectrometer is assessed by monitoring its instrument line shape (ILS). The ILS is determined on a routine basis by the analysis of measurements with a calibrated low-pressure HBr gas cell (HBr NDACC Cell #13 from 2002 to 2012 and Cell #48 from 2012 to present) using the LINEFIT version 14 retrieval algorithm described by Hase (1999). Until recently, the ILS has been monitored using the InSb detector and an internal source, so the light path differs slightly from that of the solar beam used in atmospheric measurements. Since 2019, ILS measurements have been made using the same optical path as the solar measurements, using either the sun or a globar as the source. The ILS is determined as follows: first a reference spectrum is recorded at a resolution of 0.004 cm$^{-1}$. Then the HBr gas cell is placed in the beam path

inside the evacuated sample compartment and a second spectrum is recorded. The transmission spectrum is calculated by taking the ratio of the HBr spectrum and the reference spectrum, thus eliminating any systematic features. This transmission spectrum is fitted to optimize the modulation efficiency and the phase error as a function of optical path distance (OPD) until the residuals are minimized.

The quality of the ILS is described by its modulation efficiency and phase error. The time series of retrieved modulation efficiency, along with the value at the maximum OPD of 250 cm are plotted Figure 2. The LINEFIT retrieval algorithm assumes the modulation efficiency at 0 cm is 1, and so values can be larger than 1. The modulation efficiency curves differ significantly, indicating variable instrument alignment. In general, the modulation efficiency has improved in recent years due to efforts to re-align the DA8, however there are two periods with particularly low modulation efficiency with values at
maximum OPD near 0.4: mid-2009 and mid-2012 to mid-2013. These low modulation efficiency values, which correspond to an ILS that differs significantly from the ideal, have not had a notable effect on the retrieved total columns and profiles for any of the species presented in this work, save for the period between December 6, 2007 and March 30, 2009 which is discussed below. Unfortunately, no ILS measurements were made during this period. Attempts to model the modulation efficiency by including it as a fitting parameter in the retrieval have resulted in unphysical values for the modulation efficiency, even when
highly constrained, and no notable improvement in the retrievals.

More recently, with fine adjustments to the optical configuration of the FTIR spectrometer, the modulation efficiency has been maintained closer to 1, yet there are curves where over-modulation occurs near 60 cm, indicating a minor shear error in the alignment. The phase error curves, shown in Figure 3, generally show an initial decrease in value, then increase to a peak before decreasing again. Most phase error curves vary between 0 and 0.06 with a few outliers. Since mid-2013, the phase
error has been fairly consistent, which along with the improved modulation efficiency, suggests that alignment issues have been resolved. A recent instrument line shape measurement is shown in Figure 3.

The measurements from December 6, 2007 to March 30, 2009 are affected by an instrument artifact, which appears to be due to instrument misalignment and gives rise to anomalous profile retrievals that are particularly notable for $CH_4$. Total columns can be adjusted by adding an interpolated offset as in Bader et al. (2017), however profiles remain atypical. Efforts to correct
for this misalignment during the retrievals have not been successful, and so $CH_4$ retrievals for this period are no longer archived and retrievals of other gases for this period should be used with caution.

## 2.2 Heliostat

In 2001, an altitude-azimuth heliostat (manufactured by AIM Controls Inc.) was installed on the roof above the FTIR laboratory to direct the solar beam to the DA8 spectrometer. The heliostat's mobile mirror rotates in both azimuth and altitude (elevation)
to follow the sun, directing the solar beam to a fixed mirror that reflects it into the laboratory, to two flat folding mirrors, a 90° off-axis parabolic mirror, and finally through the filter wheel and into the entrance aperture of the FTIR spectrometer as shown in Figure 1(a). The AIM Controls heliostat operated in both active and passive solar tracking modes from 2002 to 2014. In the



passive mode, the heliostat used ephemeris calculations to track the position of the sun. In the active mode, four photodiodes were used to keep the solar beam centered. This active tracking system had reliability and accuracy issues and over time it

increasingly required operator intervention to ensure the stability of the solar beam alignment on the entrance aperture of the FTIR spectrometer. The nominal pointing accuracy, determined by the distribution of altitude and azimuth corrections that must be applied to the moving mirror to keep the solar beam aligned, provided by AIM Controls was better than 60 arcsec. However, this pointing accuracy degenerated over time and by 2014 it had degraded to approximately 200 arcsec (0.056°). Figure 4(a) shows the distribution of corrections for one day of measurements using this quadrant diode system; different

clusters are the result of manually adjusting the alignment after the solar beam drifted or was obstructed by clouds. Figure 4(b) shows the distribution of corrections at one position.

On September 2, 2014, a new active tracking system was implemented: the Community Solar Tracker (CST, Franklin, 2015). Instead of four photodiodes, a Charge Coupled Device (CCD) camera intercepts a portion of the solar beam and the solar image is used for tracking. It has been shown that such a system can achieve better pointing accuracy than a quadrant diode

(Gisi et al., 2011). The CST active tracking software was adapted to the optics of TAO. Images of the sun are processed with an ellipse-fitting algorithm that determines the center position of the solar disc. With this information, it is possible to calculate the azimuth and altitude corrections necessary to maintain the center of the disk at the desired position to keep the solar beam aligned. The resulting pointing accuracy, as seen in Figure 5, is better than ~30 arcsec (Yamanouchi, 2021), a significant improvement on the previous active tracking system. Direct comparison of the photodiode system and the active CST solar

tracker is not possible due to differences in modes of operation. However, the standard deviation (1σ) in solar line shifts (calculated from CO retrievals) decreased from $3.99 \times 10^{-3}$ cm$^{-1}$ (2002-2014) to $1.96 \times 10^{-3}$ cm$^{-1}$ (2014-2019 after switching to the camera solar-disk-fitting system (see Figure 6) (Yamanouchi, 2021). Solar line shifts are caused by the tracker failing to point at the center of the sun, resulting in Doppler shifting of the incoming solar radiation, and thus provide a measure of tracking performance.

**2.3 Complementary instruments at TAO**

In addition to the FTIR spectrometer, TAO hosts several instruments which record complementary measurements on a regular basis. Local meteorological parameters, such as temperature, humidity, wind speed and direction, precipitation, and solar irradiance are measured by a commercially available weather station (Vantage Pro Plus manufactured by Davis Instruments Corp.). Current and previously recorded conditions are publicly available

(http://www.atmosp.physics.utoronto.ca/wstat/index.htm). In addition, temperature and pressure measurements (with nominal accuracies of ±0.2 °C and ±0.15 hPa respectively) have been recorded by a PTU300/PTB330 Class A temperature and pressure sensor manufactured by Vaisala since March 2013. Ozone total columns were measured by an Environment and Climate Change Canada (ECCC) Brewer spectrophotometer from March 2005 to January 2016 these measurements can be accessed through the World Ozone and Ultraviolet Data Centre (http://www.woudc.org). The Brewer spectrophotometer was replaced




by an ECCC Pandora UV-visible spectrometer in July 2016; these data are available through the Pandonia Global Network ([http://data.pandonia-global-network.org/](http://data.pandonia-global-network.org/), instrument number 109, station "St. George"). An open-path FTIR system was installed in November 2017 and has been running nearly continuously since then (Byrne et al., 2020; You et al., 2021). Finally, solar irradiance is measured by a Kipp & Zonen CM3 pyranometer, operated by ECCC.

TAO has hosted additional instruments for shorter periods: an ECCC Picarro gas analyzer was used to measure ambient levels

of O, $CO_2$, $CH_4$ and water vapour from November 2013 to January 2015 and a Very High Frequency (VHF) lightning detector was in use between 2008 and 2010 (Abreu et al., 2010). Other instruments have been deployed intermittently between field campaigns, including a Systeme d'Analyse par Observations Zenithales (SAOZ, [http://saoz.obs.uvsq.fr/](http://saoz.obs.uvsq.fr/)) and the University of Toronto Ground-Based Spectrometer (UT-GBS) which uses differential optical spectroscopy to retrieve $O_3$, $NO_2$ and BrO (Fraser, 2008).

**3 Data processing**

**3.1 Retrieval strategy**

Total and partial columns of $O_3$, HCl, HF, $HNO_3$, $CH_4$, $C_2H_6$, CO, HCN, $N_2O$, $C_2H_2$, $H_2CO$, $CH_3OH$, HCOOH and $NH_3$ are retrieved from the TAO FTIR solar absorption measurements using the optimal estimation method (OEM) as formulated in Rodgers (1976, 1990, 2000) and implemented in the SFIT4 v0.9.4.4 retrieval code

([https://wiki.ucar.edu/display/sfit4/Infrared+Working+Group+Retrieval+Code,+SFIT](https://wiki.ucar.edu/display/sfit4/Infrared+Working+Group+Retrieval+Code,+SFIT)).

SFIT4 couples OEM with a non-linear Newtonian iteration scheme to optimize the spectral fit in the given microwindows based on the HIgh-resolution TRANsmission (HITRAN) 2008 spectral line list (Rothman et al., 2009). $CH_4$ retrievals have been studied extensively by Sussmann et al. (2011), who recommended the Tikhonov regularization approach (Tikhonov, 1963; Twomey, 1963), instead of OEM. The TAO $CH_4$ retrievals follow the approach outlined in that work with the tuning parameter,

α, optimized to yield a value of 2 for the median degrees of freedom for signal (DOFS, defined as the trace of the averaging kernel matrix, Rodgers (2000), and discussed in more detail below). $N_2O$ is also retrieved using the Tikhonov regularization approach to minimize oscillations in the retrieved profile (Angelbratt et al., 2011), however in the case of $N_2O$, α is optimized for a median DOFS of 3.

Temperature and pressure profiles used in the retrieval are obtained from the US National Centers for Environmental

Prediction (NCEP) meteorological data product (calculated for each NDACC site and available at [https://www-air.larc.nasa.gov/missions/ndacc/data.html?NCEP=ncep-list](https://www-air.larc.nasa.gov/missions/ndacc/data.html?NCEP=ncep-list)).

The early TAO FTIR measurements were evaluated during a number of ground-based and satellite measurement comparisons (Wiacek et al., 2007; Wunch et al., 2007; Taylor et al., 2007, 2008). Some retrieval parameters have since been updated, such as the *a priori* profiles, which are now based on the mean profile of a 40-year Whole Atmosphere Community Climate Model





(WACCM) run (Marsh et al., 2013). The exception to this is $NH_3$; the *a priori* profile for $NH_3$ was based on the *a priori* profile used at the NDACC site in Bremen (Dammers et al., 2015), which was obtained from balloon-based measurements (Toon et al., 1999). The *a priori* profiles for each species are shown in red, along with all retrieved profiles from 2002 to 2020 in grey, on both linear and log scales in Figure 7.

The spectral microwindows have also been updated as recommended by the NDACC IRWG harmonization initiative. Typical
measured (Obs) and modeled (Calc) spectra, along with the contributions of the primary and interfering gases, are shown in Figures 8 to 21. Efforts were made to reduce systematic features in the residuals; however, in some cases this was not entirely successful. For both species retrieved using Tikhonov regularization ($CH_4$ and $N_2O$), it is expected that there may be features in the residuals because of the smoothness constraint imposed on the retrieved profile as part of this approach. Consequently, the peaks of the lines are typically not well fitted for either of these species. Additionally the solar lines in the CO
microwindows and one of the $O_3$ lines in the second CO microwindow are not well fitted in the majority of the spectra; however, other than increasing the root mean square (RMS) spectral fit error, these features have little effect on the retrieved CO profile.

Table 2 lists the microwindows, interfering gases, retrieval parameters (descriptions of the *a priori* and measurement covariance matrices) and the resulting mean DOFS for the total columns and the sensitivity range of the retrieved profiles over the entire dataset.

All retrieved profiles for the NDACC stratospheric species ($O_3$, HCl, HF, $HNO_3$), the NDACC tropospheric species ($CH_4$, $C_2H_6$, CO, HCN, $N_2O$), and the research products ($C_2H_2$, $H_2CO$, $CH_3OH$, HCOOH, $NH_3$) are plotted in grey in Figure 7. The profiles are retrieved on a 48-layer grid that spans from 0 to 120 km as prescribed by NDACC. The impact of vertical discretization of the atmosphere on modeled transmission spectra in the mid-infrared range was previously investigated by Wiacek and Strong (2008), who found that while systematic differences in modeled transmissions can be seen in most gases
retrieved at TAO, some gases (such as $O_3$) are more affected by vertical discretization than others.

The retrieval starts with the creation of a model atmosphere using meteorological information and *a priori* profiles of the target species, and interfering species, if any. The initial spectra for the selected microwindows (see Table 1) are calculated using the forward model, which encapsulates the physics of the measurement, i.e., the relationship at each height between gas concentration, optical depth, and transmission. This is calculated on a line-by-line basis assuming a Voigt lineshape and
accounting for instrumental lineshape effects. Then, the profile of the trace gas of interest is iteratively adjusted; at each iteration step, the profile to be retrieved is adjusted, and the spectrum is calculated and compared to the measured spectrum. This iterative process is repeated until the calculated and measured spectra converge. Convergence criteria are determined by examining the chi-squared ($\chi^2$) value of the root mean square (RMS) of the difference between the calculated and measured spectra.

The signal-to-noise ratio of the measurements (SNR) is the ratio of the maximum signal in the selected microwindow to the noise level for each spectrum (the standard deviation of the zero-level signal in a microwindow) and provides a measure of the



quality of the spectrum. The SNR is used to define the diagonal elements of the measurement covariance matrix, $S_\varepsilon$, i.e., $S_{\varepsilon ii} = 1/SNR^2$ (Wiacek et al., 2007). SNR is calculated individually for each of the filters discussed in Section 2.1, and the spectral windows used to calculate the SNR for each filter are listed in Table 1. The mean SNR for the set of microwindows used for

the retrievals for each species is included in Table 2.

The *a priori* covariance matrix, $S_a$ can be directly specified in the input files for running SFIT4. This matrix should capture the natural variability in the profile of the retrieved species and can be used as a tuning parameter of the retrieval. $S_a$ is typically a diagonal matrix, with the diagonal elements defined as the expected spread in the trace gas profile for that particular region of the atmosphere. It is usually expressed as a percentage value. In some cases, it also includes off-diagonal elements that are

determined by the interlayer correlation length. The interlayer correlation is modeled as an exponential function with half-width equal to the correlation length and is included in Table 2. A correlation length of 4 km is typically used in these retrievals, and is an average value derived from the HALogen Occulatation Experiement (HALOE) climatology (Russell et al, 1993; Wiacek, 2007). Off-diagonal elements are added to *a priori* covariance matrix primarily for tropospheric species or in cases for which the inclusion of these elements results in notable improvement to the smoothness of the retrieved profile, without

affecting total column values.

The averaging kernels characterize the sensitivity of the retrieved profile to the true profile and provide information about the vertical resolution of the retrievals (Rodgers, 2000). The averaging kernels, the rows of the averaging kernel matrix, A, are plotted for all fourteen species in Figure 22. In the top row of Figure 22, the volume mixing ratio averaging kernels are plotted in units of ppmv / ppmv. The averaging kernels can also be used to define the independent partial columns that can be derived

from the FTIR measurements. The trace of the averaging kernel matrix is referred to as the degrees of freedom for signal, and determines the number of independently resolved pieces of information present in retrieved profile. The mean DOFS for all retrieved profiles from 2002 to 2020 for each species are listed in the seventh column of Table 2.

In the bottom row of Figure 22, the so-called "sensitivity" profiles (Vigouroux et al., 2008) and the total column averaging kernels are plotted in units of molecules $cm^{-2}$ / molecules $cm^{-2}$. The sensitivity at a given height is the sum of the elements in

the row of the corresponding averaging kernel. In the OEM formulation, the sensitivity describes the balance between information provided by the measurement and information derived from the *a priori*. A value of 1 indicates that the retrieved state is completely independent of the *a priori*, while a value of 0 indicates the opposite. An ideal sensitivity profile would be 1 at all heights. The sensitivity profiles for the species retrieved at TAO all show regions of independence from the *a priori* profile. Regions where the sensitivity is greater than 0.5 for each species are listed in the final column of Table 2. In these

regions, more than 50% of the retrieved profile information comes from the measurement.

**3.2 Error analysis**

The uncertainties in each retrieved atmospheric profile are determined following the methodology presented in Rodgers (1976; 1990). The measurement error ($S_m$) and forward model errors ($S_f$) are calculated as described in those references:





$$S_m = GS_\epsilon G \tag{1}$$

$$S_f = GK_b S_b K_b^T G^T \tag{2}$$

where $S_\epsilon$ is the previously introduced measurement covariance matrix, $K_b$ is Jacobian matrix for the vector of forward model parameters, $b$, and $S_b$ is the covariance error matrix for $b$. $G$ is the gain matrix:

$$G = (S_a^{-1} + K^T S_\epsilon^{-1} K)^{-1} K^T S_\epsilon \tag{3}$$

where $S_a$ is the *a priori* covariance matrix previously alluded to, and $K$ is the weighting function matrix that relates the measurement state to the true state, i.e., the forward model discussed previously.

The forward model errors considered here are the solar zenith angle error ($S_{sza}$), the line parameter errors ($S_{line}$), and the temperature profile errors. For each forward model error, a covariance matrix is created to describe the best estimate of uncertainty in the parameter. These forward model covariance matrices are discussed below. The interference error ($S_{int}$), described in Rodgers and Connor (2003) as the error that accounts for uncertainties in the simultaneously retrieved interfering gases and retrieval parameters ($S_{ret}$) such as wavelength shift, background slope, solar line shift and phase error, is included in our error calculation.

The smoothing error ($S_s$), sometimes called the null-space error, is not included in the error calculation as per the recommendation of the NDACC IRWG. The smoothing error describes the error induced by using a finite layering grid and thus is considered to be intrinsic with this method of retrieval (von Clarmann, 2014). Data users can calculate the smoothing error as appropriate for their application using the following equation:

$$S_s = (A - I)S_x(A - I)^T \tag{4}$$

where $A$ is the averaging kernel, archived with every retrieval, $I$ is the identity matrix and $S_x$ is the covariance matrix of the true atmospheric state around the mean state. $S_x$ can be estimated based on climatology.

The errors are divided into two types: systematic ($S_{sys}$) and random errors ($S_{ran}$). Measurement error, solar zenith angle error, retrieval parameter error, and the interference error are considered to be random and the line parameter errors are considered to be systematic. The temperature error has both a random ($S_{temp\_r}$) and systematic ($S_{temp\_s}$) component. The total systematic and random errors are determined by adding all components in quadrature, and both the covariance matrix and total column uncertainty (in molecules cm$^{-2}$) for each of these components of the total error are included in the archived HDF files:

$$S_{sys} = S_{line} + S_{temp\_s} \tag{5}$$

$$S_{ran} = S_{sza} + S_{ret} + S_{int} + S_{temp\_r}. \tag{6}$$

The solar zenith angle (SZA) uncertainty relates to the calculation of the airmass, as the SZA describes the line of sight of the instrument and thus the path of the light through the atmosphere. For the airmass calculation, it is assumed that the solar



tracker is pointing directly at the center of the sun. There are three aspects to this error: the uncertainty in the SZA calculation (which is based on the measurement time), the uncertainty in the pointing of the solar tracker, and the change in the SZA over

the scan time of the measurement. To achieve higher SNR, four spectra are averaged (co-added), resulting in a total scan time of 20 minutes at 0.5 cm/s (scanning speed from 2001 to March 7, 2014) and 10 minutes at 1 cm/s (scanning speed from March 8, 2014 to 2016). In 20 minutes, the average change in SZA throughout the entire year at the Toronto site is 0.86° and in 10 minutes the average change is 0.43°. The uncertainty caused by the variation in the SZA outweighs any uncertainty caused by the two other aspects of this error. Thus a variance of 0.86° is used for the solar zenith angle uncertainty from 2001 to March

7, 2014 and 0.43° from March 8, 2014 to 2020.

Three values describe the line parameters in the forward model, each with its own uncertainty: the line intensity, the temperature-broadened half-width, and the pressure-broadened half-width. The uncertainties in the respective covariance matrices are based on the errors provided in HITRAN. When an uncertainty range is given in HITRAN 2008 (Rothman et al., 2009), the average of the range is used, and when no value is provided, an uncertainty of 20% is assumed. The line parameter

uncertainties for each species are listed in Table 3. For CO, CH$_4$, and O$_3$, the uncertainty analyses were done using a consolidated methodology (standardized by NCAR; https://github.com/NCAR/sfit-processing-environment, last accessed: May 2021) to meet the standards set for contributing to the CAMS RD system.

As mentioned above, the temperature error has both a systematic and a random component, thus requiring both systematic and random covariance matrices. The uncertainty in the NCEP temperature profile used in the retrieval is determined by comparing

all the radiosonde measurements for a given year to the daily NCEP profile at Toronto corresponding to the radiosonde launch date. The average difference is taken as the systematic uncertainty and the standard deviation of the differences as the random uncertainty. The nearest radiosonde site to TAO is the Buffalo Niagara International airport (42.56°N, 78.44°W, 215 m.a.s.l.), 145 km away and across Lake Ontario, and the measurements compared are for the year 2010. Radiosonde temperature measurements accuracies decrease above 30 km (Luers and Eskridge, 1995), therefore we use the mean value for the systematic

error from 0.55 to 29.14 km for the systematic error and the NCEP error for the random error above this height. The calculated values at 0.174 km were replaced with mean uncertainties from 0.55 to 29.14 km in an attempt to mitigate the effect of the difference in location between the radiosonde measurement and the NCEP profile. The resulting systematic and random forward model temperature uncertainties. can be found in Table 4.

Vertical profiles of the mean systematic and random errors from 2019 are shown in Figure 23 for each gas, along with profiles

of the individual error terms as described above. For some species, one of the sources of error clearly dominates, such as the SZA uncertainty for CH$_4$, while for others, such as O$_3$, all the sources of error are of the same order of magnitude. Since the error profiles presented here are based solely on the diagonal variance terms of the error covariance matrices, it is most useful to view them qualitatively: they can be used to get a sense of the magnitude of the errors, the scope of the different uncertainty sources, and the height regimes that are most uncertain. An interlayer correlation length is used to model the off-diagonal

elements of the error matrices, as is the case for the *a priori* covariance matrix. The full error covariance matrices (one for



systematic errors and one for random errors) are provided in the HDF file for each measurement and should be used to calculate the estimated errors for the total and partial columns.

### 3.3 Quality assurance

Quality assurance is the final step in the data processing. Any filtering is a trade-off between maintaining data quality and
rejecting measurements that could be indicative of atmospheric processes that occur infrequently or are associated with particular events such as biomass burning events and stratospheric ozone intrusions. For each gas in this dataset, a threshold is defined to remove measurements that are exceptional in terms of large root-mean-square (RMS) spectral fit error or small DOFS. A large RMS may indicate that the retrieval scheme failed to capture the shape of the spectrum, resulting in an erroneous total column, or may simply indicate a particularly noisy spectrum. Smaller than average DOFS are, in most cases,
associated with low quality measurements and thus should be rejected. To filter based on both of these criteria, we apply a quality threshold based on the ratio of the RMS to the DOFS such that retrievals for which the RMS/DOFS ratio is greater than the threshold are discarded. Additional advantages of such a filter are detailed in Sussmann et al. (2011).

For each species, a series of threshold values were tested and a value was selected which removed results with large RMS residuals and/or small DOFS and with anomalous total columns that did not correlate with other results either from the species
in question or other species; the chosen values can be found in Table 5. For $CH_4$, CO, and $O_3$, RMS/DOFS filtering is not performed, but further quality control is done to meet the standards set by CAMS (submitted files must pass an automated quality assurance filter, which examines retrieved parameters, including the retrieved column profile, water profile, and averaging kernels). For $H_2CO$, the RMS/DOFS filtering is still used.

### 4 Time series

This section presents the archived time series for the species routinely retrieved from TAO FTIR solar spectra, following the sequence of NDACC stratospheric species, NDACC tropospheric species, and research products. Outliers in all the time series were examined but were not rejected unless there were issues with the spectral fit or the retrieved profile was deemed unphysical. The information content of the total columns and tropospheric and stratospheric partial columns (i.e., the mean DOFS and systematic and tropospheric errors) from 2002 to 2020 are listed in Table 6.

Datasets are archived as HDF (Hierarchical Data Format; https://www.hdfgroup.org/, last accessed: May 2021) files. For each species, the files are separated by year (typically, datasets are archived annually). If improvements are made to the retrieval, datasets may be updated on the NDACC archival server and version number of the file incremented.





### 4.1 Ozone

The TAO total column time series of ozone ($O_3$) from 2002 to 2020, shown in Figure 24(a), is dominated by the stratospheric
partial column. The stratospheric partial column contains the majority of the information, with average degrees of freedom
for signal (DOFS) of 3.16, but there is also sensitivity in the troposphere, with DOFS of 1.19. The mean DOFS for the total
column is 4.38 and the mean systematic and random error are 5.32 % and 1.52 % respectively. The tropospheric measurements
have been examined by Whaley et al. (2015), who analysed $O_3$ air pollution events in Toronto to identify their local, regional
and long-range sources using the GEOS-Chem model. The seasonal cycle in Figure 24(b) shows the relative monthly mean
from 2002 to 2020, calculated by normalising the monthly means by the respective annual mean over the 2002-2020 dataset,
e.g., January 2003 data are normalized by the annual mean from 2003, and so on, and all January mean values are then
averaged. The observed seasonal cycle exhibits a spring maximum when transport is dominant, and a fall minimum when
photochemical loss dominates. The Brewer-Dobson circulation transports $O_3$ produced in the tropics to the mid- to high-
latitudes, where it descends (Butchard, 2014), increasing mid-latitude $O_3$ over the course of the winter, while photochemical
ozone loss reduces $O_3$ during the summer. Outliers in the $O_3$ total columns, seen in Figure 24(c), are likely signatures of
dynamic variability over Toronto, some of which were attributed to polar vortex intrusions by Whaley et al. (2013).

### 4.2 Hydrogen chloride

The main source of hydrogen chloride (HCl) in the middle atmosphere is the photodissociation of chlorofluorocarbons (CFCs)
and hydrochlorofluorocarbons (HCFCs). HCl is a chlorine reservoir species and plays a role in the catalytic ozone destruction
cycle. The stratospheric partial columns, shown in Figure 25(a)(red), are almost the same as the total columns (blue), having
near identical mean DOFS (1.76 and 1.78), while the tropospheric columns contain no information (DOFS of 0). The mean
systematic and stratospheric errors are also nearly identical between the total and stratospheric columns, 2.06 % and 2.04 %
for the mean systematic error and 1.58 % and 1.57 % for the mean random error, which is dominated by the SZA error. The
seasonal cycle, seen most clearly in Figure 25(b), is driven by tropopause height, with a minimum when the tropopause height
is at its highest, in summer, and a maximum when the tropopause height is at its lowest, in winter (Kohlhepp et al., 2012). The
time series was examined in Kohlhepp et al. (2012). The variability of the HCl measurements, seen Figure 25(c), has been
investigated and many of these enhancement events correspond to intrusions of polar vortex air, as studied by Whaley et al.
(2013).

### 4.2 Hydrogen fluoride

Much like hydrogen chloride, hydrogen fluoride (HF) is a stratospheric species which has photodissociation of CFCs and
HCFCs as its source, and thus HF is a good indicator of anthropogenic activity (Kohlhepp et al., 2012). The FTIR measurement
has no sensitivity to the troposphere (mean DOFS of 0.01), thus the stratospheric partial column measurements, plotted in red
in Figure 26(a), are almost identical to the total column measurements. Consequently, the total columns and stratospheric



columns have similar mean DOFS (1.85 and 1.83). The total column systematic error is predominantly due to temperature, with a mean error of 2.43 %, and the total column random error has similar contributions from all sources with a mean of 1.95 %. The seasonal cycle is also driven by the tropopause height (Kohlhepp et al, 2012). Since HF has a lifetime exceeding 10 years, the variability in the measurements is indicative of stratospheric dynamics. Enhancements of HF can be used to identify polar intrusion events as in Whaley et al. (2013), with events typically occurring at least twice per year. The TAO HF measurement time series has previously been examined in Kohlhepp et al. (2012) who discussed the decrease in HCl and $ClONO_2$ emissions from 2000-2009 due to the restrictions of anthropogenic chlorine source gases under the Montreal Protocol, and the increase in HF as fluorine emissions are not explicitly restricted. A recent study using TAO data from 2002 to 2019 found a positive trend in the HF total columns at a rate of 0.59 ± 0.11 %/year (Yamanouchi et al., 2021a).

### 4.4 Nitric acid

Nitric acid ($HNO_3$) is a reservoir species for nitrogen in the stratosphere and plays an important role in stratospheric ozone chemistry by sequestering reactive $NO_x$ radicals, thus reducing the destruction of ozone (e.g., Jacob, 1999). This sequestration occurs when $HNO_3$ is produced via the oxidation of $NO_x$ by OH. $HNO_3$ is ultimately converted back to $NO_x$ by reactions with OH or by photolysis. $HNO_3$ can be removed from the stratosphere by transport to the troposphere where it is scavenged by precipitation. The $HNO_3$ total column time series is shown in blue in Figure 27(a) and the mean DOFS is 1.27. The mean systematic and random errors for the $HNO_3$ total column retrieval are 0.37 % and 3.42 % respectively; the systematic error is dominated by the spectroscopic error, while the largest component of the random error is the measurement error. The TAO FTIR measurements are predominantly sensitive to the stratosphere (red) with mean DOFS of 1.07, while the mean DOFS in the troposphere are 0.21. $HNO_3$ has a seasonal cycle, seen in Figure 27(b), with a winter maximum and a summer minimum. The variability in the $HNO_3$ measurements increases in winter as seen in Figure 27(c), in agreement with model predictions for a mid-latitude site presented in Toohey et al. (2007).

### 4.5 Methane

The time series of methane ($CH_4$) over Toronto is shown in Figure 28(a). The FTIR measurements have DOFS of ~1 in both the troposphere and the stratosphere (mean DOFS of 1.09 and 1.06, respectively) and the total mean DOFS is 2.11. The mean systematic error is on the total columns is 3.92 %, and is primarily due to the temperature error. The mean random error on the total columns is lower, at 1.99 %, and is dominated by the SZA error. $CH_4$ is the most abundant hydrocarbon in the atmosphere (IPCC, 2013) and is long-lived with a lifetime of between 9 and 12 years. It is lost at a faster rate in the summer due to oxidation by OH. Due to its many sources and sinks, the amplitude of the $CH_4$ seasonal cycle is small and is not visible in the TAO total column time series, indicated by the flat mean relative monthly mean time series shown in Figure 28(b). The TAO $CH_4$ time series has been examined by Bader et al. (2017), who describe the recent increase in $CH_4$ during the 2005-2014 period. Yamanouchi et al. (2021a) found a positive trend of 0.41 ± 0.03 %/year in the TAO $CH_4$ total column from 2009 to 2019.





### 4.6 Ethane

Ethane ($C_2H_6$) is moderately long-lived in the atmosphere, with a lifetime of about two months (Rudolph, 1995). The time series of TAO $C_2H_6$ total columns is shown in blue in Figure 29(a), along with the stratospheric (red) and tropospheric (green) partial columns. The sensitivity of the TAO $C_2H_6$ FTIR measurements is mostly tropospheric (mean DOFS of 1.85); however, there is some sensitivity in the lower stratosphere (mean DOFS of 0.39 in the stratosphere) which is difficult to discern in the figure as the stratospheric partial columns are significantly smaller than the total and tropospheric columns. Consequently, the mean systematic error (4.36 %) and random error (2.10 %) on the total columns are primarily from the troposphere. The seasonal cycle is driven by OH oxidation and thus a winter maximum and summer minimum can be seen in the plot of the mean relative monthly means in Figure 29(b). TAO measurements of $C_2H_6$ have been used to study the effects of oil and natural gas extraction on ethane and methane emissions in North America (Franco et al., 2016). Isolated $C_2H_6$ events, such as the one used to identify a biomass burning plume that passed over Toronto on 21 July 2011 during the 2011 summer BORTAS campaign (Griffin et al., 2013), can be seen in Figure 29(c), where all the measurements for all years have been plotted from January to December and are colour-coded by year. The BORTAS event is partly obscured by comparable total column values of $C_2H_6$ in 2014.

### 4.7 Carbon monoxide

The time series of carbon monoxide (CO) total columns, shown in Figure 30(a), is dominated by the troposphere, where CO concentrations are largest, thus the tropospheric column measurements differ only slightly from the total column. The FTIR CO retrieval does have some sensitivity in the stratosphere (mean DOFS of 0.39) but the sensitivity is largely tropospheric (mean DOFS of 1.18) and the mean DOFS of the total column is 2.16. This sensitivity in the troposphere makes CO a useful indicator for biomass burning events (Griffin et al., 2013). The total column random error, with a mean of 1.66 %, is principally composed of the line intensity error. The mean systematic error on the total columns is 2.46 %. The main sink of CO in the atmosphere, as with $CH_4$ and $C_2H_6$, is oxidation by OH. Consequently, the seasonal cycle is similar to those of $CH_4$ and $C_2H_6$, but because CO is a by-product of hydrocarbon oxidation, the summer minimum is slightly delayed as seen in Figure 30(b). A secondary source of CO is fossil fuel combustion. The TAO CO time series has been used to study urban ozone pollution events (Whaley et al., 2015), to evaluate the Community Atmosphere Model with Chemistry (CAM-Chem) assimilation of Measurement of the Pollution in the Troposphere (MOPITT) measurements (Gaubert et al., 2016), and to validate MOPITT data (Buchholz et al., 2017). TAO CO columns have been used to identify biomass burning plumes transported overhead (Lutsch et al., 2016, 2020; Yamanouchi et al., 2020).

### 4.8 Hydrogen cyanide

Hydrogen cyanide is mainly emitted by biomass burning, although urban emission from vehicles (Baum et al., 2007) is also significant. The FTIR retrieval has sensitivity in the troposphere and stratosphere, with mean DOFS of 0.89 and 1.19,





respectively, and the mean DOFS of the total column is 2.09. The HCN time series is shown in Figure 31(a). The concentrations of HCN are much higher in the troposphere, so the tropospheric measurements, shown in green, dominate the total column. The total column mean random error is 5.23 % and is predominantly measurement error. The total column mean

systematic error is 3.88 % and is dominated by the temperature error. The seasonal cycle, shown in Figure 31(b), is characterized by a peak in summer of varying amplitude, which is consistent with the variable biomass burning source, and a winter minimum. Individual enhancement events, seen in the time series in Figure 31(c), are indicators of biomass burning events. The large enhancement in summer 2014 was due to transport from the Northwest Territories fires as shown by Lutsch et al. (2016).

**4.9 Nitrous oxide**

Nitrous oxide ($N_2O$) is the fourth most important anthropogenic greenhouse gas (IPCC, 2013), but thus far emissions of $N_2O$ are not controlled. $N_2O$ has many sources: the ocean, soils, biomass burning, chemical fertilizers, and livestock (Jacob, 1999). With the phasing out of CFCs under the Montreal Protocol and its subsequent amendments, it has become the most important ozone-depleting substance (Ravishankara et al., 2009). $N_2O$ is long-lived, with a lifetime of approximately 114 years. It is

450 transported up to the stratosphere where it is lost through photodissociation and oxidation to $N_2$ or oxidation to NO, contributing to the loss of $O_3$ by the $NO_x$-catalyzed mechanism (Bates and Hays, 1967; Crutzen, 1970; McElroy and McConnell, 1971). The FTIR measurements have sensitivity both in the stratosphere (mean DOFS of 1.24) and the troposphere (mean DOFS of 1.66), with the total and partial column time series plotted in Figure 32(a). The mean total column DOFS is 2.90. For the total columns, the mean systematic error is 3.59 % and is dominated by the temperature error, while the mean random

error is 1.44 %, which is principally composed of the SZA error. The TAO total column measurements of $N_2O$ show no seasonal cycle, but a seasonal cycle does exist in the stratospheric partial columns, shown in Figure 32(b) and has been examined in Whaley et al. (2013). Diminished total columns values, shown Figure 32(c), have been used to identify mid-latitude polar vortex intrusions in Whaley et al. (2013).

**4.10 Acetylene**

The main sources of acetylene ($C_2H_2$) are combustion of fossil fuels, biofuel use and biomass burning (Xiao et al., 2007 and references therein). The total columns of $C_2H_2$ measured at TAO, shown in Figure 33(a), are dominated by the tropospheric partial columns, with mean total DOFS of 1.42 and mean tropospheric DOFS of 1.34. The total column mean random error is 34.04 %, which is primarily the measurement error. The systematic error on the total columns is smaller, with a mean of 10.88 %, principally temperature error. Unlike CO and HCN, biomass burning emissions of $C_2H_2$ are generally not observed

in the time series for Toronto. Although the lifetime of $C_2H_2$, approximately two weeks (Xiao et al., 2007), is sufficiently long to support transport of biomass burning emissions, its abundance during these events is low and difficult to detect within the $C_2H_2$ total columns at Toronto, which are dominated by anthropogenic sources that account for most of the variability seen in Figure 33(c). The seasonal cycle of $C_2H_2$, shown in Figure 33(b), has a minimum in the summer months as a result of its





destruction by oxidation by OH. Maximum total columns are observed in the winter months when production of OH is minimal
due to decreased ozone photolysis.

## 4.11 Formaldehyde

Formaldehyde (H₂CO) is produced by oxidation of CH₄ and volatile organic compounds (VOCs). It reacts rapidly with OH
and has a lifetime of several hours. It is an important species for understanding photo-oxidation pathways of the atmosphere
(Jones et al., 2009).   The total column of H₂CO (mean DOFS of 1.23) is almost entirely located in the troposphere (mean
DOFS of 1.07) as shown in Figure 34(a).  The mean systematic error on the total columns is 13.66 %, principally composed
of line intensity error, and the total column mean random error is 7.71 %, primarily composed of SZA error. The seasonal
cycle of H₂CO at Toronto, shown in Figure 34(b), is mainly driven by its production by oxidation of CH₄ and VOCs, which
are emitted by plants and vegetation in the summer months, as well as by fossil fuels. H₂CO concentrations are highly variable,
due to its short lifetime, ranging from total columns of $10^{15}$ to $10^{16}$ molecules cm$^{-2}$, as can be seen in Figure 34(c).

## 4.12 Methanol

Methanol (CH₃OH) is one of the most abundant non-methane volatile organic gases in the atmosphere (Jacob et al., 2005) and
is an important precursor for the formation of CO and H₂CO (Hu et al., 2009 and references therein). Methanol is relatively
short-lived with a lifetime ranging from several days in the boundary layer (Heikes et al., 2002) to several weeks on a global
scale (Jacob et al., 2005; Stavrakou et al., 2011). The total columns of CH₃OH, shown in Figure 35(a), are dominated by the
tropospheric partial column.  Accordingly, the mean DOFS of the total column is 1.49 and the mean DOFS of the tropospheric
partial column is only slightly lower at 1.41.  For total columns, the mean systematic error is 15.31 % and the mean random
error is 3.72 %.  The seasonal cycle at Toronto, plotted in Figure 35(b), shows increasing total columns from the spring to
summer months, falling below the detection limit in winter. This seasonal cycle is due to natural and biogenic sources of
CH₃OH, while the main loss is due to reaction with OH.

## 4.13 Formic acid

Formic acid (HCOOH) is produced mainly by photochemical oxidation of volatile organic compounds (Millet et al., 2015;
Franco et al., 2021) and is the second most abundant global organic acid in the atmosphere (Zander et al., 2010). It is also
emitted directly by vegetation (Kesselmeier and Staudt, 1999; Kesselmeier, 2001) and biomass burning (Good et al. 2000).
The main sinks of formic acid are its reaction with OH and both dry and wet deposition (Stavrakou et al., 2012). The
atmospheric lifetime of HCOOH is relatively short and has been estimated to be on the order of three or four days (Paulot et
al., 2009). The total columns of HCOOH at Toronto (mean DOFS of 1.06), shown in Figure 36(a), are mostly tropospheric
(mean DOFS of 0.95).  For total columns, the mean systematic error is 10.41 % and the mean random error is 9.18 %. The
high percentage mean errors are due to errors on small total column amounts.  The random error is dominated by the retrieval
parameters error and the systematic error is dominated by the pressure-broadening error. The seasonal cycle of HCOOH at

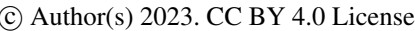

Toronto, shown in Figure 36(b), is driven by biogenic emissions as observed by the sharp increase in the total columns during the summer months. Due to the short lifetime of HCOOH, long-range transport of biomass burning emissions are likely not an important source of HCOOH at Toronto, making attribution of enhancements, seen in the time series plot in Figure 36(c), to biomass burning events difficult (Yamanouchi et al., 2020).

**4.14 Ammonia**

Ammonia ($NH_3$) is highly reactive and soluble, and it is the most abundant alkaline trace gas. It reacts rapidly with other trace gases to contribute to aerosol formation (Hertel et al., 2012) and therefore has notable effects on air quality (Pope et al. 2009; Erisman et al., 2011). The main sources of $NH_3$ are agricultural emissions, biogenic sources and fossil fuel use. Biomass burning also contributes to considerable emissions of $NH_3$ (e.g., Bouwman et al., 1997). The FTIR time series of $NH_3$, shown in Figure 37(a), is only sensitive to the boundary layer and troposphere where $NH_3$ is most abundant. The mean DOFS for

$NH_3$, which is primarily contained within the tropospheric partial column, is 1.10. The total column mean systematic error is 11.85 % and the total column mean random error is 13.25 %. The systematic error is primarily measurement error and the random error is primarily temperature error. Similarly to HCCOH, these high percentage errors reflect the small total columns of $NH_3$ that occur throughout the time series. The time series of $NH_3$ total columns at Toronto show a weak seasonal cycle, as seen in in Figure 37(b). Yamanouchi et al. (2021b) have shown that TAO total column $NH_3$ has similar seasonality to surface

$NH_3$ measurements. Although $NH_3$ columns are largest in the summer months, its emissions are episodic as seen in Figure 37(c). The short lifetime of ammonia, ranging from one to five days (Warneck, 1998), also contributes to its variability. Biomass burning is an important source of $NH_3$, but its short lifetime makes long-range transport of $NH_3$ to Toronto infrequent. However, in August 2014, enhancements of $NH_3$ along with simultaneous enhancements of CO, HCN and $C_2H_6$ were observed and attributed to fires in the Northwest Territories, as discussed by Lutsch et al. (2016). Toronto $NH_3$ measurements have also

been used to validate the Infrared Atmospheric Sounding Interferometer (IASI) $NH_3$ product (Dammers et al., 2016) and CrIS (Cross-track Infrared Sounder) $NH_3$ retrieval column and profile measurements (Dammers et al., 2017).

**5 Summary**

This work describes the 19-year trace gas time series derived from high-resolution FTIR solar absorption spectra recorded at the University of Toronto Atmospheric Observatory. The instrumentation is described, along with the retrieval strategy, error

analysis, and quality assurance methods. The following nine trace gases are retrieved regularly as part of the NDACC IRWG: $O_3$, HCl, HF, $HNO_3$, $CH_4$, $C_2H_6$, CO, HCN and $N_2O$. Additionally, $C_2H_2$, $H_2CO$, $CH_3OH$, HCOOH and $NH_3$ are retrieved routinely. Total column measurements, retrieved profiles, averaging kernels, error values, and error covariance matrices, along with retrieved or *a priori* water vapour columns and profiles depending on the retrieval strategy are included in the GEOMS HDF data files that are archived on NDACC data repository and on Borealis, the Canadian Dataverse Repository. In addition

to the total column time series, partial column time series can be calculated based on the information provided, allowing users

to study the different layers of the atmosphere. These data products have been included in a number of studies since routine measurements began in 2002 and can continue to be used to gain a better understanding of the dynamics, chemistry and evolution of the atmosphere over Toronto, Ontario, Canada.

**Data availability**

The TAO FTIR data are regularly archived on the NDACC data repository at https://www-air.larc.nasa.gov/missions/ndacc/data.html?station=toronto.tao/hdf/ftir/, and the data presented in this work are available on Borealis, the Canadian Dataverse Repository at https://doi.org/10.5683/SP2/VC8JMC (Yamanouchi et al., 2022).

**Code availability**

The SFIT4 retrieval code is available at https://wiki.ucar.edu/display/sfit4 (Hannigan et al., 2022).

**Author contribution**

SY, SC, and KS conceived this study. AW, JT, CHW, SC, SY, EL, SR, and OC operated the TAO FTIR and contributed to data acquisition and earlier analyses. SY performed the SFIT4 retrievals, following from previous work by SC and EL. SC wrote an early version of the paper, and SY wrote the final version. All of the authors discussed the results, and reviewed and provided commentary on the manuscript. KS was responsible for funding acquisition and management of TAO.

**Competing interests**

The authors declare that they have no conflict of interest.

**Acknowledgements**

Funding for this work was provided by the Natural Sciences and Engineering Research Council of Canada (including the NSERC CREATE Training Program in Technologies for Exo-Planetary Science), the Canadian Space Agency, and
550 Environment and Climate Change Canada. The TAO measurements have been supported in the past by the Canadian Foundation for Climate and Atmospheric Sciences, ABB Bomem Inc., the Canada Foundation for Innovation, the Ontario Research and Development Challenge Fund, the Premier's Research Excellence Award, and the University of Toronto. The CST active tracking software was adapted to work with TAO optics by Yi Gao, O. Colebatch, and S. Roche. We also wish to thank the many students, postdocs, and interns who have contributed to TAO data acquisition over the last two decades.



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

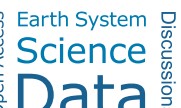

Table 1: NDACC IRWG band pass filter set used at TAO. All filters were manufactured by Northumbria Optical Coatings Limited.

| NDACC filter | Wavenumber range [cm⁻¹] | Wavelength range [µm] | Target species | Serial number | Start date | Wedge angle [arc min] | Type of filter | Window used for SNR calculation [cm⁻¹] |
|---|---|---|---|---|---|---|---|---|
| 1 | 4000 - 4300 | 2.3 - 2.5 | HF | TFBP/555/00N | 18/10/2001 | $8 \pm 3$ | Band Pass | 4007.50 - 4007.64, |
| | | | | BP-3495-11-4-07 | 20/10/2013 | $10 \pm 1$ | Band Pass | 4038.727 - 4038.871 |
| | | | | BP-4428-1-10-11 | 30/07/2014 | $10 \pm 1$ | Band Pass | |
| 2 | 2900 - 3500 | 2.6 - 3.3 | HCN, $C_2H_2$ | TFBP/556/00N | 13/02/2002 | $8 \pm 3$ | Band Pass | 3381.275 - 3381.536, |
| | | | | BP-3495-2-5-07 | 10/06/2010 | $10 \pm 1$ | Wide Band Pass | 3485.65 - 3485.77 |
| 3 | 2400 - 3100 | 3.3 - 4.1 | HCl, $CH_4$, $C_2H_6$, $N_2O$, $H_2CO$ | TFBP/557/00N | 18/10/2001 | $8 \pm 3$ | Band Pass | 2402.20 - 2402.30, 2924.866 - 2925.100 |
| 4 | 2000 - 2700 | 3.9 - 5.0 | CO | TFBP/558/00N | 22/10/2001 | $8 \pm 3$ | Band Pass | 2300.00 - 2300.50, 2526.228 - 2526.618 |
| 5 | 1500 - 2200 | 4.7 - 6.3 | CO | TFBP/559/00N | 10/12/2002 | $8 \pm 3$ | Long Wave Pass | 1985.260 - 1985.510 |
| | | | | BP-3495-5-5-07 | 20/07/2009 | $10 \pm 1$ | Wide Band Pass | |
| 6 | 750 - 1350 | 7.4 - 1.4 | $O_3$, $HNO_3$, $CH_3OH$, HCOOH, $NH_3$ | TFLWP/560/00N | 07/01/2002 | $8 \pm 3$ | Long Wave Pass | 700.00 - 700.50, 1139.075 - 1139.168 |
| | | | | LWP-3495-6-4-07 | 24/05/2013 | $10 \pm 1$ | Long Wave Pass | |



Table 2: Summary of fitting parameters for TAO FTIR retrievals. Mean DOFS are for the entire 2002-2020 dataset, and mean SNR is calculated with one year of data from 2019. The version numbers in the first column indicate the accompanying version of the data archived on Borealis and currently archived on the NDACC data repository.

| Species | Microwindows [cm⁻¹] | Interfering species | $S_a$ [%] | Interlayer correlation length [km] | Mean SNR ± 1σ | Mean DOFS ± 1σ | Sensitivity range (sensitivity > 0.5) [km] |
|---|---|---|---|---|---|---|---|
| O₃ (v4) | 782.56 - 782.86<br>788.85 - 789.37<br>993.30 - 993.80<br>1000.0 - 1004.5 | $H_2O$, $CO_2$, $O_3^{668}$, $O_3^{686}$<br>$H_2O$, $CO_2$, $O_3^{668}$, $O_3^{686}$<br>$H_2O$, $CO_2$, $O_3^{668}$, $O_3^{686}$, $C_2H_4$<br>$H_2O$, $CO_2$, $O_3^{668}$, $O_3^{686}$, $C_2H_4$ | 10 | None | 120 ± 43 | 4.38 ± 0.56 | 0.36 - 46.68 |
| HCl (v2) | 2727.73 - 2727.83<br>2775.70 - 2775.80<br>2925.80 - 2926.00 | $O_3$, HDO<br>$O_3$, $N_2O$<br>$O_3$, $NO_2$, CH4 | 10 | None | 994 ± 479 | 1.78 ± 0.36 | 14.3 - 40.2 |
| HF (v2) | 4038.81 - 4039.07 | $H_2O$, $CH_4$ | 50 | None | 346 ± 155 | 1.85 ± 0.34 | 18.9 - 40.2 |
| HNO₃ (v2) | 867.05 - 870.00 | $H_2O$, OCS, $NH_3$ | 20 | 4 | 116 ± 41 | 1.27 ± 0.38 | 1.66 - 26.9 |
| CH₄ (v4) | 2613.70 - 2615.40<br>2835.50 - 2835.80<br>2921.00 - 2921.60 | HDO, $CO_2$<br>HDO<br>HDO, $H_2O$, $NO_2$ | Tikhonov Reg. | n/a | 1002 ± 450 | 2.11 ± 0.25 | 0.36 – 29.9 |
| C₂H₆ (v2) | 2976.66 - 2976.95<br>2983.20 - 2983.55<br>2986.50 - 2986.95 | $H_2O$, $O_3$, $CH_4$<br>$H_2O$, $O_3$, $CH_4$<br>$2O$, $O_3$, $CH_4$ | 30 | 4 | 594 ± 227 | 1.85 ± 0.31 | 0.36 - 15.4 |
| CO (v5) | 2057.70 - 2058.00<br>2069.56 - 2069.76<br>2157.50 - 2159.15 | $O_3$, $CO_2$, OCS<br>$O_3$, $CO_2$, OCS<br>$O_3$, $CO_2$, OCS, $N_2O$, $H_2O$ | 20 | 4 | 592 ± 165 | 2.16 ± 0.29 | 0.36 - 35.3 |
| HCN (v2) | 3268.05 - 3268.40<br>3287.10 - 3287.35<br>3299.40 - 3299.60<br>3331.40 - 3331.80 | $H_2O$, $C_2H_2$<br>$CO_2$, $C_2H_2$<br>$H_2O$, $H_2^{18}O$<br>$H_2O$, $H_2^{17}O$, $CO_2$, $N_2O$ | 20 | 4 | 444 ± 150 | 2.09 ± 0.50 | 2.2 - 29.9 |
| N₂O (v3) | 2481.30 - 2482.60<br>2526.40 - 2528.20<br>2537.85 - 2538.80<br>2540.10 - 2540.70 | None | Tikhonov Reg. | n/a | 864 ± 276 | 2.90 ± 0.33 | 0.36 - 28.4 |
| C₂H₂ (v2) | 3250.43 - 3250.77<br>3255.18 - 3255.725 | $H_2O$<br>$H_2O$ | 100 | 4 | 275 ± 84 | 1.42 ± 0.33 | 0.36 - 16.5 |



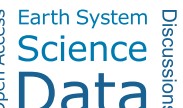

| Species | Wavenumber range | Interfering species | | | | | |
|---|---|---|---|---|---|---|---|
| H$_2$O (v3) | 3304.825 - 3305.35 | H$_2$O, HDO | | | | | |
| H$_2$CO (v3) | 2778.12 - 2778.80 | CH$_4$, CO$_2$, O$_3$, N$_2$O | 100 | 4 | 299 ± 107 | 1.23 ± 0.16 | 0.36 - 28.4 |
| | 2780.60 - 2781.17 | CH$_4$, CO$_2$, O$_3$, N$_2$O | | | | | |
| CH$_3$OH (v2) | 992.00 - 998.70 | O$_3$, O$_3^{686}$, O$_3^{668}$, O$_3^{676}$, H$_2$O, CO$_2$ | 100 | 4 | 143 ± 29 | 1.49 ± 0.19 | 0.36 - 15.4 |
| | 1029.00 - 1037.00 | O$_3$, O$_3^{686}$, O$_3^{668}$, O$_3^{676}$, O$_3^{667}$, H$_2$O, CO$_2$ | | | | | |
| HCOOH (v3) | 1102.00 - 1109.00 | O$_3$, H$_2$O, CCl$_2$F$_2$, CHF$_2$Cl, NH$_3$, HDO, N$_2$O, CH$_4$ | Tikhonov Reg. | n/a | 149 ± 54 | 1.06 ± 0.04 | 0.36 - 18.9 |
| | 1178.40 - 1178.80 | | | | | | |
| NH$_3$ (v2) | 930.32 - 931.32 | H$_2$O, O$_3$, CO$_2^1$, CO$_2^2$, N$_2$O, CO$_2^3$, HNO$_3$ | 50 | 4 | 160 ± 55 | 1.10 ± 0.11 | 0.36 - 22.8 |
| | 966.97 - 967.675 | H$_2$O, O$_3$, CO$_2^1$, CO$_2^2$, N$_2$O, CO$_2^3$, HNO$_3$ | | | | | |





**Table 3: HITRAN 2008 parameter uncertainties used for the error analysis (Rothman et al., 2009), for all archived species. For CO, CH₄, and O₃, the retrievals were done using the CAMS RD retrieval methodology (see Section 3.2 for more details).**

| Species | Line intensity uncertainty (%) | Line pressure-broadening uncertainty (%) | Line temperature-broadening uncertainty (%) |
|---|---|---|---|
| †$O_3$ | 3.0 | 5.0 | 5.0 |
| HCl | 1.5 | 7.5 | 15 |
| HF | 3.2 | 1.5 | 1.5 |
| $HNO_3$ | *20 | 7.5 | 7.5 |
| †$CH_4$ | 3.0 | 5.0 | 10 |
| $C_2H_6$ | 4.0 | 4.0 | 4.0 |
| †CO | 2.0 | 5.0 | 5.0 |
| HCN | 3.5 | 3.5 | 7.5 |
| $N_2O$ | 3.5 | 3.5 | 7.5 |
| $C_2H_2$ | 15 | 3.5 | 15 |
| $H_2CO$ | 10 | 10 | 10 |
| $CH_3OH$ | 15 | *20 | *20 |
| HCOOH | 7.5 | *20 | *20 |
| $NH_3$ | 2.0 | 7.5 | 15 |

*Assumed value

†CAMS RD retrieval methodology

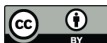



Table 4: Systematic and random forward model temperature uncertainties, derived from comparison between NCEP
daily temperature profiles at Toronto and radiosonde data from Buffalo for 2010, used for error analysis.

| Height [km] | Systematic error [K] | Random error [K] |
|---|---|---|
| 0.3507 | 1.91 | 2.56 |
| 0.7516 | 2.62 | 3.32 |
| 1.1848 | 2.57 | 3.22 |
| 1.6591 | 2.34 | 2.89 |
| 2.1697 | 1.72 | 2.16 |
| 2.7165 | 1.16 | 1.52 |
| 3.3044 | 1.11 | 1.44 |
| 3.9286 | 1.50 | 1.88 |
| 4.5890 | 1.63 | 2.02 |
| 5.2856 | 1.62 | 2.02 |
| 6.0183 | 1.52 | 1.92 |
| 6.7923 | 1.21 | 1.59 |
| 7.6025 | 1.29 | 1.78 |
| 8.4489 | 1.56 | 2.05 |
| 9.3315 | 1.49 | 2.06 |
| 10.25 | 1.50 | 1.85 |
| 11.21 | 2.13 | 2.80 |
| 12.205 | 2.45 | 3.21 |
| 13.235 | 1.53 | 2.02 |
| 14.30 | 1.66 | 1.83 |
| 15.40 | 2.31 | 2.53 |
| 16.54 | 2.50 | 2.79 |
| 17.715 | 1.88 | 1.87 |
| 18.925 | 1.54 | 1.77 |
| 20.17 | 1.45 | 1.82 |
| 21.45 | 1.58 | 1.95 |
| 22.77 | 1.37 | 1.71 |
| 24.125 | 1.47 | 1.72 |
| 25.515 | 1.57 | 1.98 |
| 26.94 | 1.96 | 2.40 |
| 28.40 | 2.17 | 2.72 |
| 29.915 | 2.75 | 3.72 |
| 31.53 | 2.00 | 5.00 |
| 33.30 | 2.00 | 5.00 |
| 35.285 | 2.00 | 6.00 |
| 37.555 | 2.00 | 6.00 |
| 40.17 | 2.00 | 7.00 |
| 43.19 | 2.00 | 7.00 |
| 46.68 | 2.00 | 7.00 |
| 50.70 | 2.00 | 9.00 |
| 55.315 | 2.00 | 9.00 |
| 60.59 | 2.00 | 9.00 |
| 66.585 | 2.00 | 9.00 |
| 73.385 | 2.00 | 9.00 |
| 81.10 | 2.00 | 9.00 |



| | | |
|---|---|---|
| 89.85 | 2.00 | 9.00 |
| 100.25 | 2.00 | 9.00 |
| 113.00 | 2.00 | 9.00 |



**Table 5:** Quality assurance thresholds applied to TAO FTIR retrievals.

| Species | RMS/DOFS threshold |
|---|---|
| $O_3$ | CAMS |
| HCl | 2.95 |
| HF | 4.5 |
| $HNO_3$ | 4.0 |
| $CH_4$ | CAMS |
| $C_2H_6$ | 4.75 |
| CO | CAMS |
| HCN | 1.4 |
| $N_2O$ | 0.3 |
| $C_2H_2$ | 7.5 |
| $H_2CO$ | 3.5 |
| $CH_3OH$ | 5.5 |
| HCOOH | 9.6 |
| $NH_3$ | 5.4 |





**Table 6: Mean total and partial column DOFS and systematic and random errors for the entire time series, as described**
**in Section 3.2.**

|  | Column | Mean DOFS $\pm 1\sigma$ | Mean systematic error [%] | Mean random error [%] |
|---|---|---|---|---|
| $O_3$ | Total | $4.38 \pm 0.56$ | 5.32 | 1.52 |
|  | Tropospheric | $1.19 \pm 0.21$ | 0.72 | 0.25 |
|  | Stratospheric | $3.16 \pm 0.36$ | 4.73 | 1.37 |
| HCl | Total | $1.78 \pm 0.36$ | 2.06 | 1.58 |
|  | Tropospheric | $0.00 \pm 0.00$ | 0.01 | 0.00 |
|  | Stratospheric | $1.76 \pm 0.35$ | 2.04 | 1.57 |
| HF | Total | $1.85 \pm 0.34$ | 2.43 | 1.95 |
|  | Tropospheric | $0.01 \pm 0.01$ | 0.00 | 0.02 |
|  | Stratospheric | $1.83 \pm 0.33$ | 2.41 | 1.93 |
| $HNO_3$ | Total | $1.27 \pm 0.38$ | 0.37 | 3.42 |
|  | Tropospheric | $0.20 \pm 0.15$ | 0.09 | 2.26 |
|  | Stratospheric | $1.07 \pm 0.23$ | 0.31 | 2.17 |
| $CH_4$ | Total | $2.11 \pm 0.25$ | 3.92 | 1.99 |
|  | Tropospheric | $1.07 \pm 0.11$ | 2.89 | 1.70 |
|  | Stratospheric | $1.04 \pm 0.14$ | 1.43 | 0.38 |
| $C_2H_6$ | Total | $1.85 \pm 0.31$ | 4.36 | 2.10 |
|  | Tropospheric | $1.46 \pm 0.22$ | 4.07 | 2.11 |
|  | Stratospheric | $0.39 \pm 0.10$ | 0.51 | 0.14 |
| CO | Total | $2.16 \pm 0.29$ | 2.46 | 1.66 |
|  | Tropospheric | $1.76 \pm 0.14$ | 2.55 | 1.64 |
|  | Stratospheric | $0.39 \pm 0.14$ | 0.53 | 0.10 |
| HCN | Total | $2.09 \pm 0.50$ | 3.88 | 5.23 |
|  | Tropospheric | $0.89 \pm 0.19$ | 2.88 | 5.51 |
|  | Stratospheric | $1.19 \pm 0.32$ | 1.12 | 0.83 |
| $N_2O$ | Total | $2.90 \pm 0.33$ | 3.59 | 1.44 |
|  | Tropospheric | $1.66 \pm 0.14$ | 2.92 | 1.22 |
|  | Stratospheric | $1.24 \pm 0.24$ | 1.20 | 0.23 |
| $C_2H_2$ | Total | $1.42 \pm 0.33$ | 10.88 | 34.04 |
|  | Tropospheric | $1.34 \pm 0.29$ | 10.94 | 34.11 |
|  | Stratospheric | $0.09 \pm 0.04$ | 0.06 | 0.40 |
| $H_2CO$ | Total | $1.23 \pm 0.16$ | 13.66 | 7.71 |
|  | Tropospheric | $1.05 \pm 0.07$ | 13.42 | 9.68 |
|  | Stratospheric | $0.18 \pm 0.09$ | 0.30 | 0.37 |
| $CH_3OH$ | Total | $1.49 \pm 0.19$ | 15.31 | 3.72 |
|  | Tropospheric | $1.41 \pm 0.16$ | 15.06 | 3.85 |





|  |  |  |  |  |
|---|---|---|---|---|
|  | Stratospheric | 0.08 ±0.03 | 0.66 | 0.50 |
| HCOOH | Total | 1.06 ± 0.04 | 10.41 | 9.18 |
|  | Tropospheric | 0.95 ± 0.02 | 9.93 | 8.79 |
|  | Stratospheric | 0.11 ± 0.02 | 0.57 | 0.69 |
| $NH_3$ | Total | 1.10 ± 0.11 | 11.85 | 13.25 |
|  | Tropospheric | 1.10 ± 0.11 | 11.85 | 13.25 |
|  | Stratospheric | 0.00 ± 0.00 | 0.00 | 0.00 |





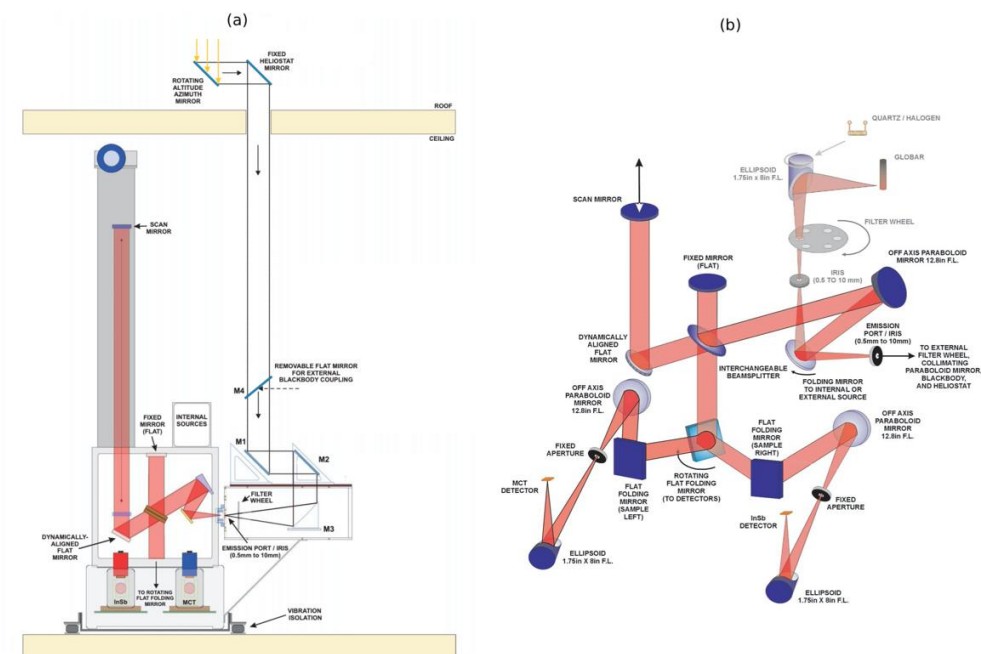

**Figure 1: (a) The FTIR spectrometer and solar tracker installation at TAO: mirrors M1, M2 and M3 couple the heliostat optics to the DA8 and M4 is a removable mirror. (b) The optical layout of the DA8 FTIR: a moving folding mirror selects between the emission port used for solar absorption measurements and internal sources (shown in grey). Figures taken from Wiacek (2006), adapted from ABB Bomem Inc. facility schematics.**

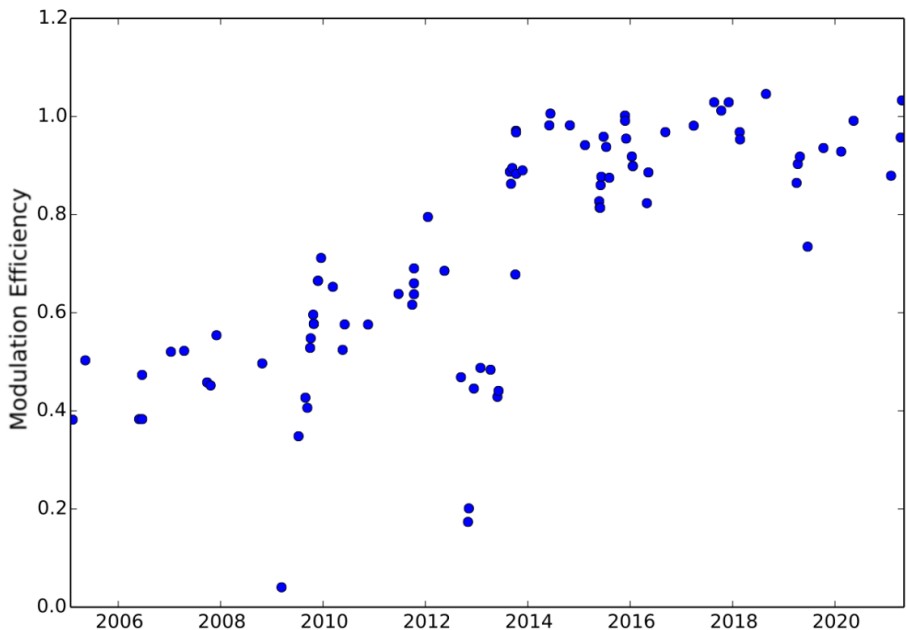

**Figure 2: Time series of TAO DA8 line shape modulation efficiency at the maximum OPD of 250 cm.**

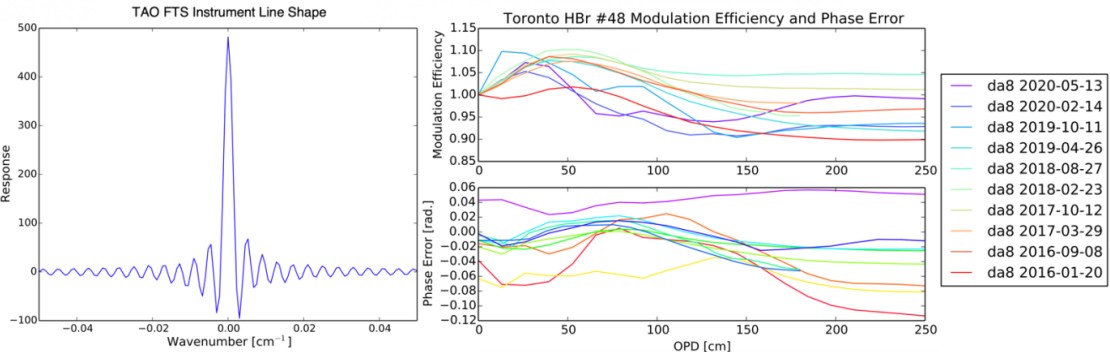

**Figure 3: TAO DA8 solar instrument line shape (left) (taken on 13 May 2020), modulation efficiency (top, right) and phase error (bottom, right) (various measurements from 2016 to 2020 shown), with HBr cell #48 at 0.004 cm$^{-1}$ resolution, obtained using LINEFIT v14.5.**





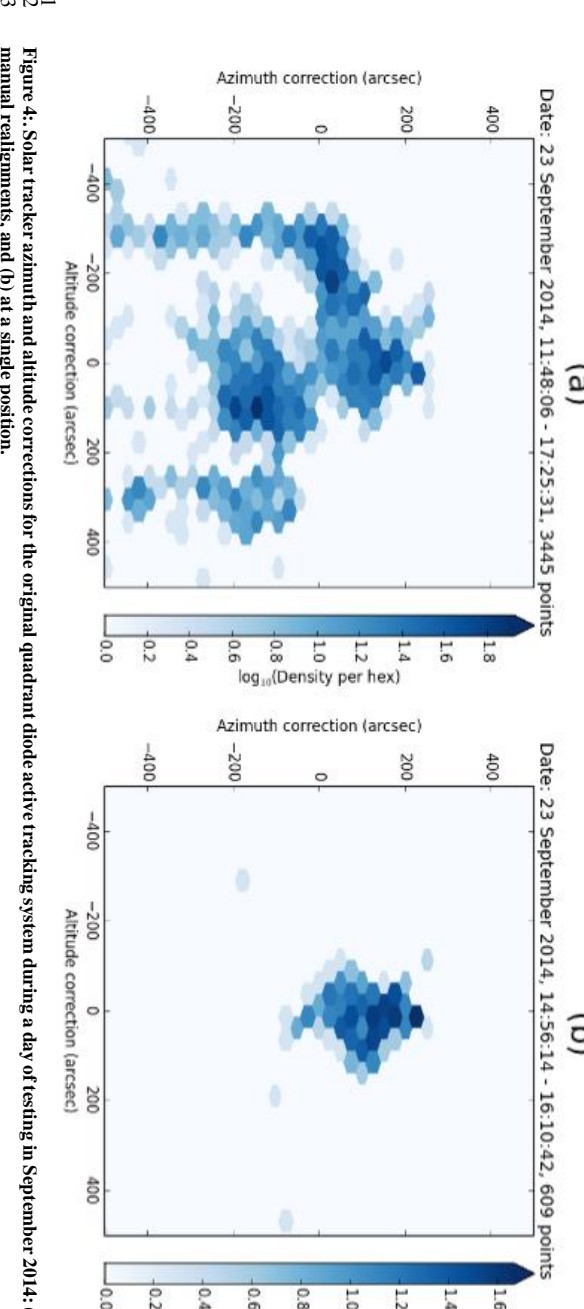

**Figure 4..Solar tracker azimuth and altitude corrections for the original quadrant diode active tracking system during a day of testing in September 2014: (a) for an entire day including manual realignments, and (b) at a single position.**

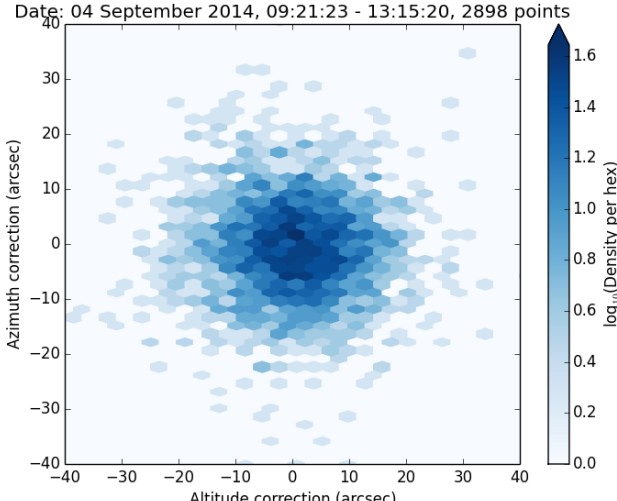

**Figure 5: Solar tracker azimuth and altitude corrections for the new CST active tracking system shortly after**
**installation in September 2014.**





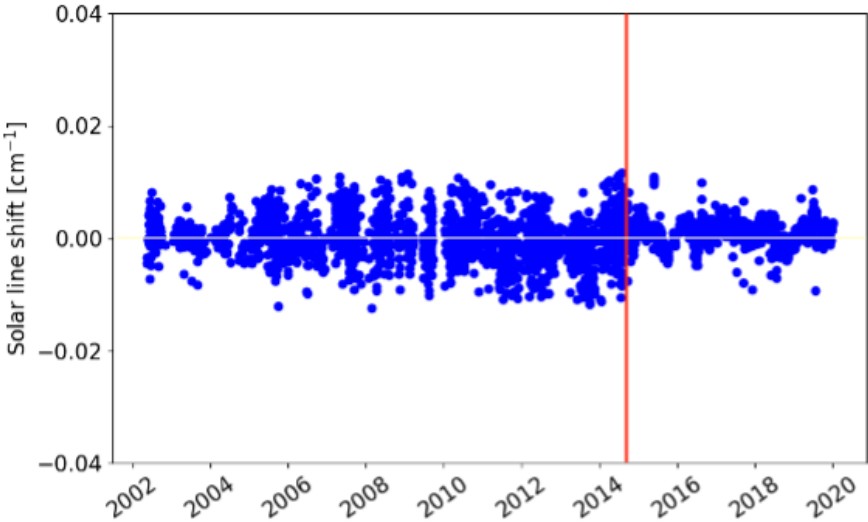

**Figure 6: Time series of the solar line shift (in cm⁻¹) from CO retrievals from 2002 to 2019. The red line indicates 2 September 2014, when the solar tracker was upgraded to the CST system. Figure taken from Yamanouchi (2021).**





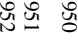

Figure 7: Retrieved profiles of O₃, HCl, HF, HNO₃, CH₄, C₂H₆, CO, HCN, N₂O, C₂H₂, H₂CO, CH₃OH, HCOOH, and NH₃ from 2002 to 2020. The mean profile is plotted in black with the two black dashed lines corresponding to ±1σ standard deviation from the mean. The *a priori* profile is plotted as a red line.

Figure 8: Sample fit (Calc) and residual (upper panels) for the O₃ fitting microwindows for a TAO FTIR transmission spectrum (Obs) recorded at 17:04:42 UTC on January 11, 2019.
The contributions of all species in the state vector are plotted separately (lower panels) and are offset to improve the clarity of the figure.


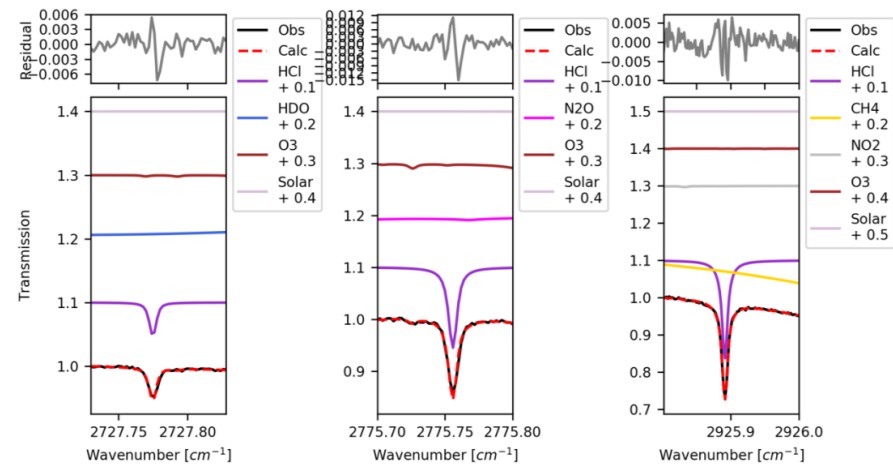

**Figure 9: Same as Figure 8 but for HCl using a spectrum recorded at 16:36:54 UTC on January 11, 2019.**



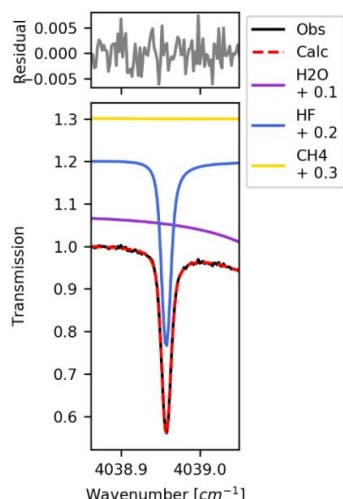

**Figure 10: Same as Figure 8 but for HF using a spectrum recorded at 17:31:50 UTC on January 11, 2019.**





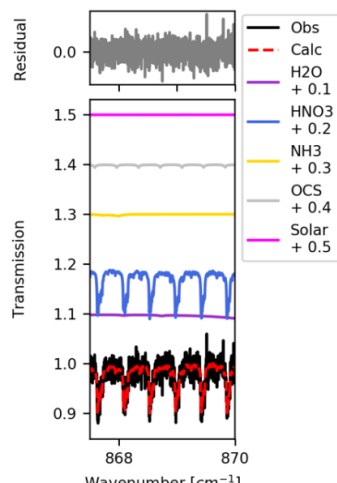

**Figure 11: Same as Figure 8 but for HNO₃ using a spectrum recorded at 19:48:59 UTC on January 11, 2019.**





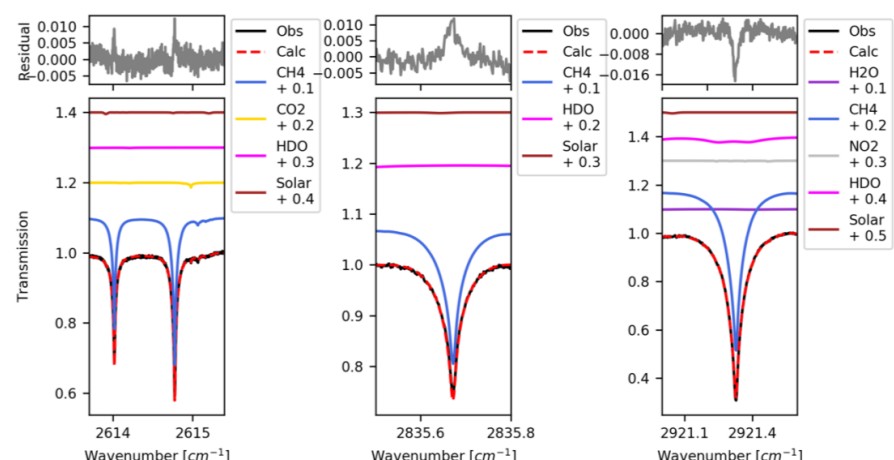

**Figure 12: Same as Figure 8 but for CH₄ using a spectrum recorded at 16:36:54 UTC on January 11, 2019.**

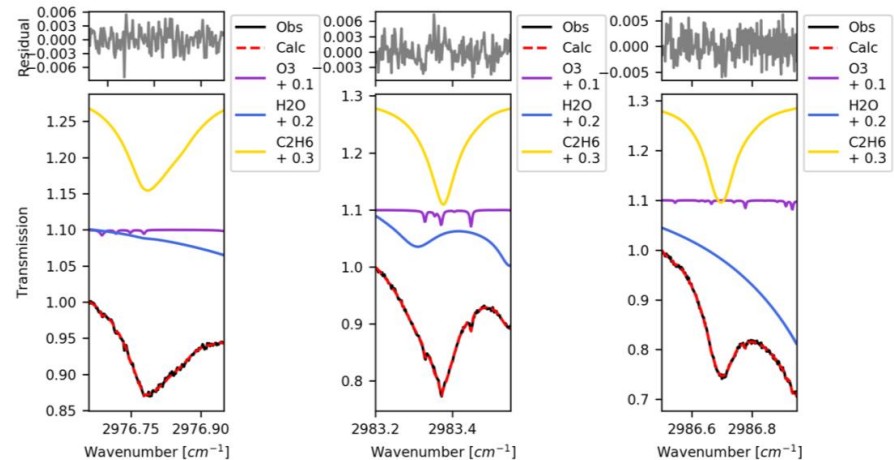

**Figure 13: Same as Figure 8 for C$_2$H$_6$, using a spectrum recorded at 16:36:54 UTC on January 11, 2019.**

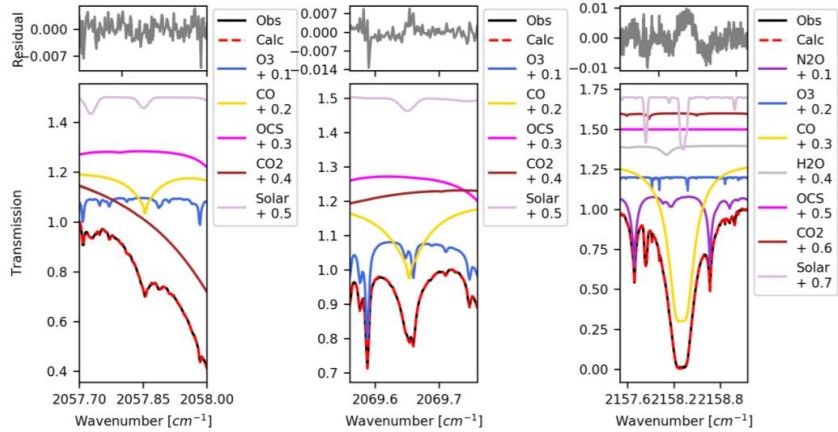

**Figure 14: Same as Figure 8 for CO, using a spectrum recorded at 16:50:37 UTC on January 11, 2019.**



Figure 15: Same as Figure 8 but for HCN, using a spectrum recorded at 16:07:45 UTC on July 15, 2019.

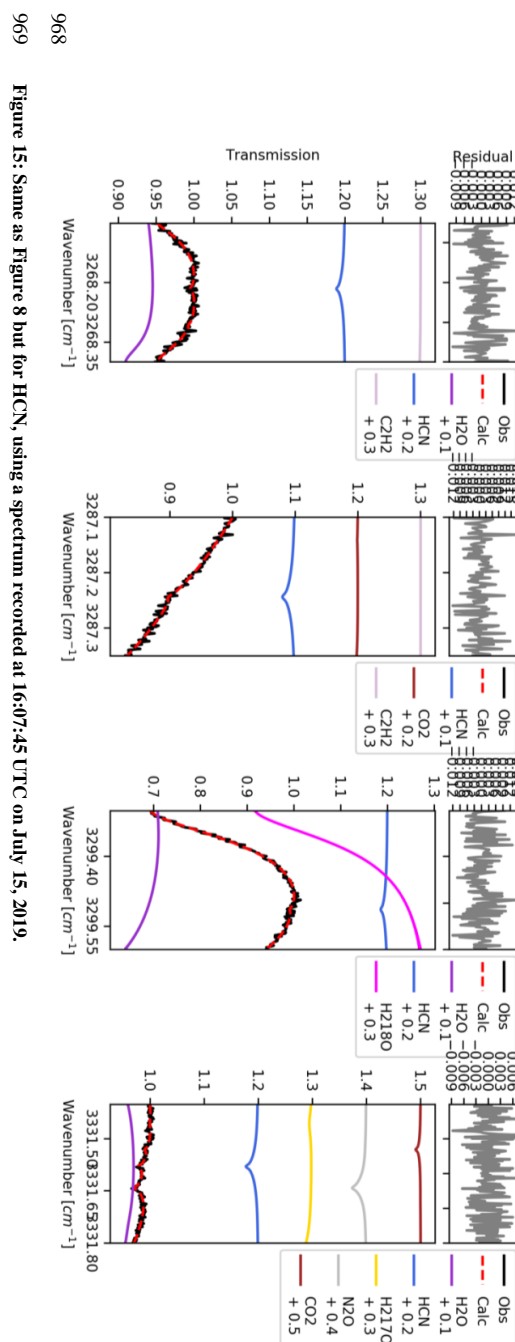





Figure 16: Same as Figure 8 but for N₂O using a spectrum recorded at 14:55:17 UTC on July 15, 2019. Since no interfering species are retrieved, solely the observed and calculated spectra are plotted.



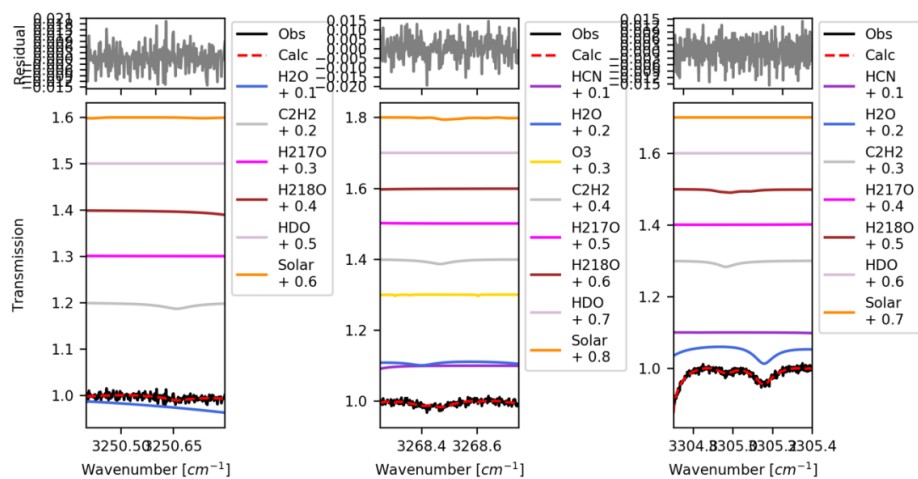

**Figure 17: Same as Figure 8 but for C₂H₂, using a spectrum recorded at 17:18:07 UTC on January 11, 2019.**





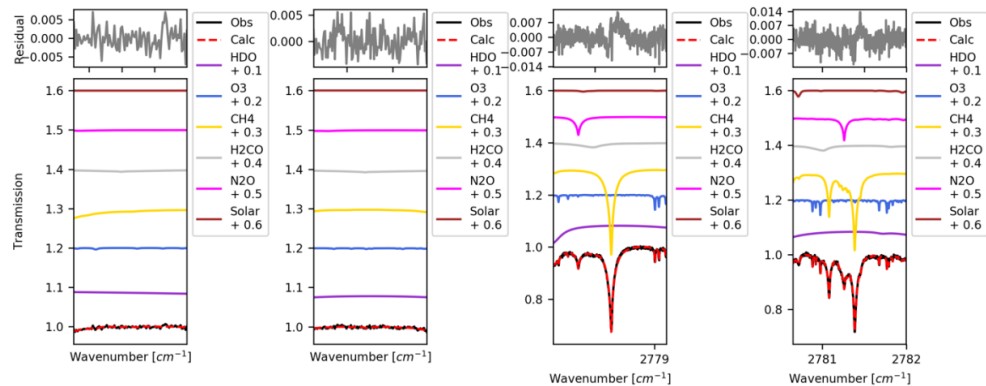

**Figure 18: Same as Figure 8 but for H₂CO, using a spectrum recoded at 15:55:17 UTC on July 15, 2093.**



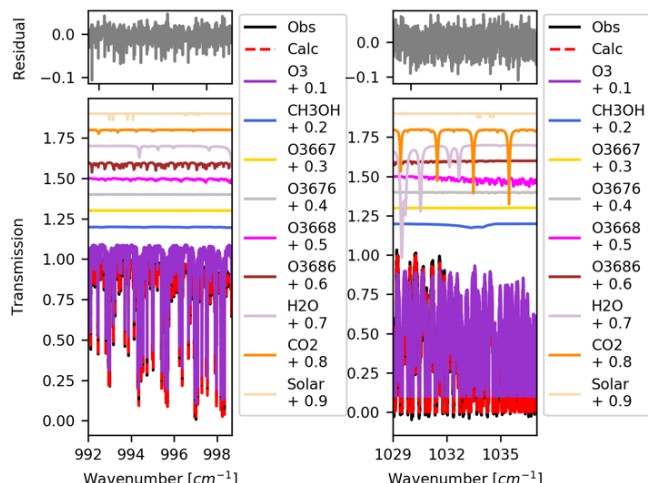

**Figure 19: Same as Figure 8 but for CH₃OH, using a spectrum recorded at 15:54:25 UTC on July 15, 2019.**



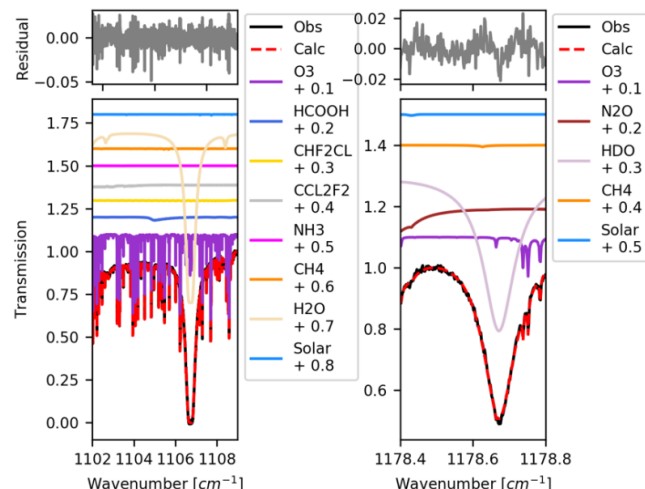

**Figure 20: Same as Figure 8 but for HCOOH, using a spectrum recorded at 15:54:25 UTC on July 15. 2019.**





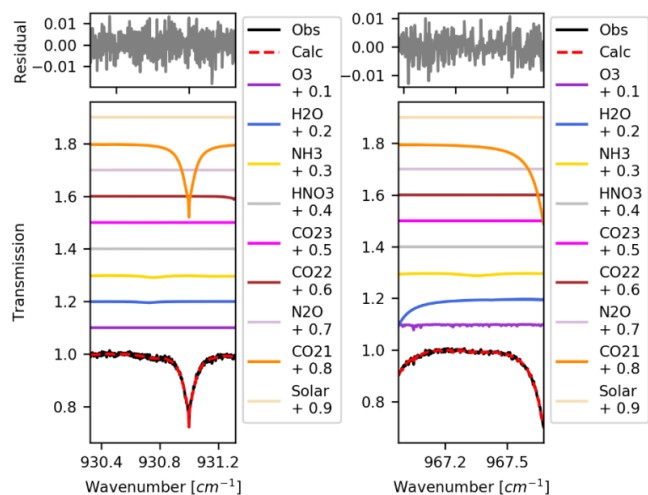

**Figure 21: Same as Figure 8 but for NH₃, using a spectrum recorded at 15:54:25 UTC on July 15, 2019.**





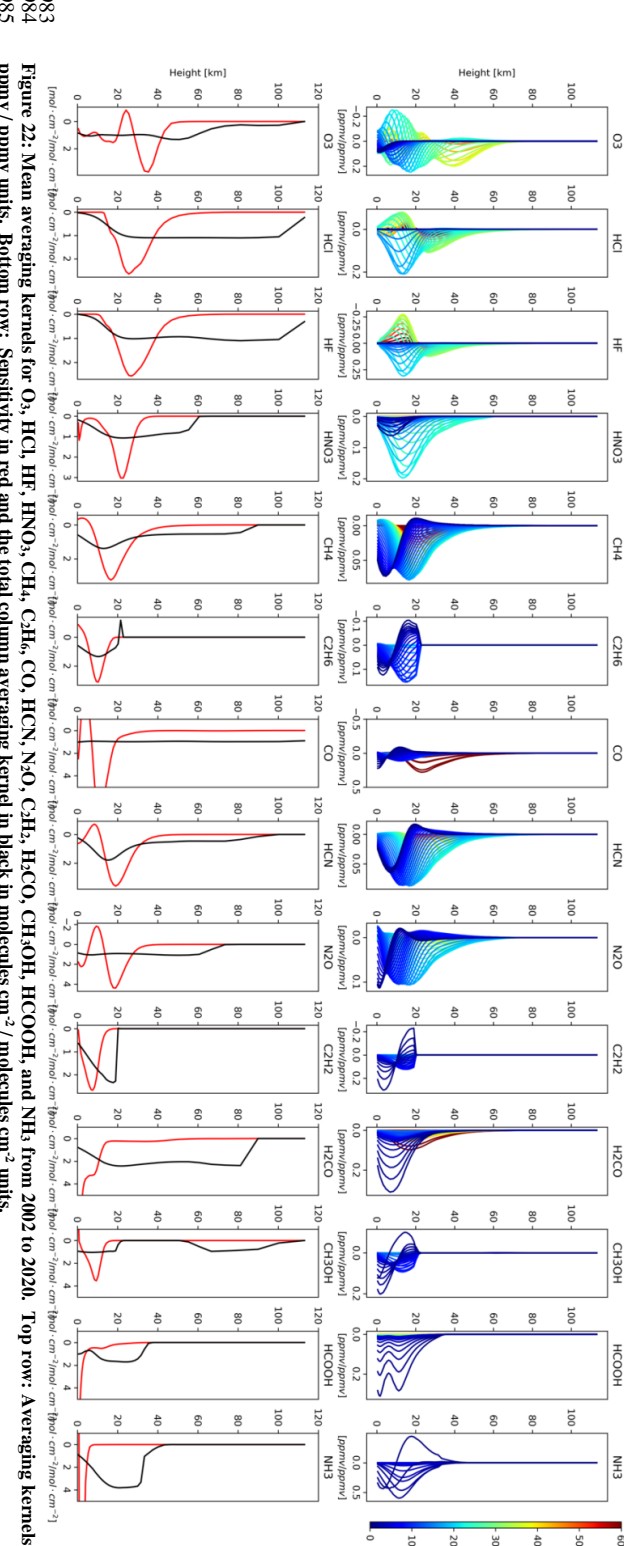

**Figure 22: Mean averaging kernels for O₃, HCl, HF, HNO₃, CH₄, C₂H₆, CO, HCN, N₂O, C₂H₂, H₂CO, CH₃OH, HCOOH, and NH₃ from 2002 to 2020. Top row: Averaging kernels in ppmv / ppmv units. Bottom row: Sensitivity in red and the total column averaging kernel in black in molecules cm⁻² / molecules cm⁻² units.**



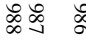

**Figure 23: Mean systematic and random error profiles (diagonal elements of $S_{sys}$ and $S_{ran}$) of $O_3$, HCl, HF, HNO$_3$, CH$_4$, C$_2$H$_6$, CO, HCN, N$_2$O, C$_2$H$_2$, H$_2$CO, CH$_3$OH, HCOOH, and NH$_3$ from 2019. Upper panels: Retrieval parameter, SZA, random temperature and measurement error comprising the random uncertainty. Lower panels: Systematic temperature, line intensity, temperature-broadening and pressure-broadening error comprising the systematic error.**

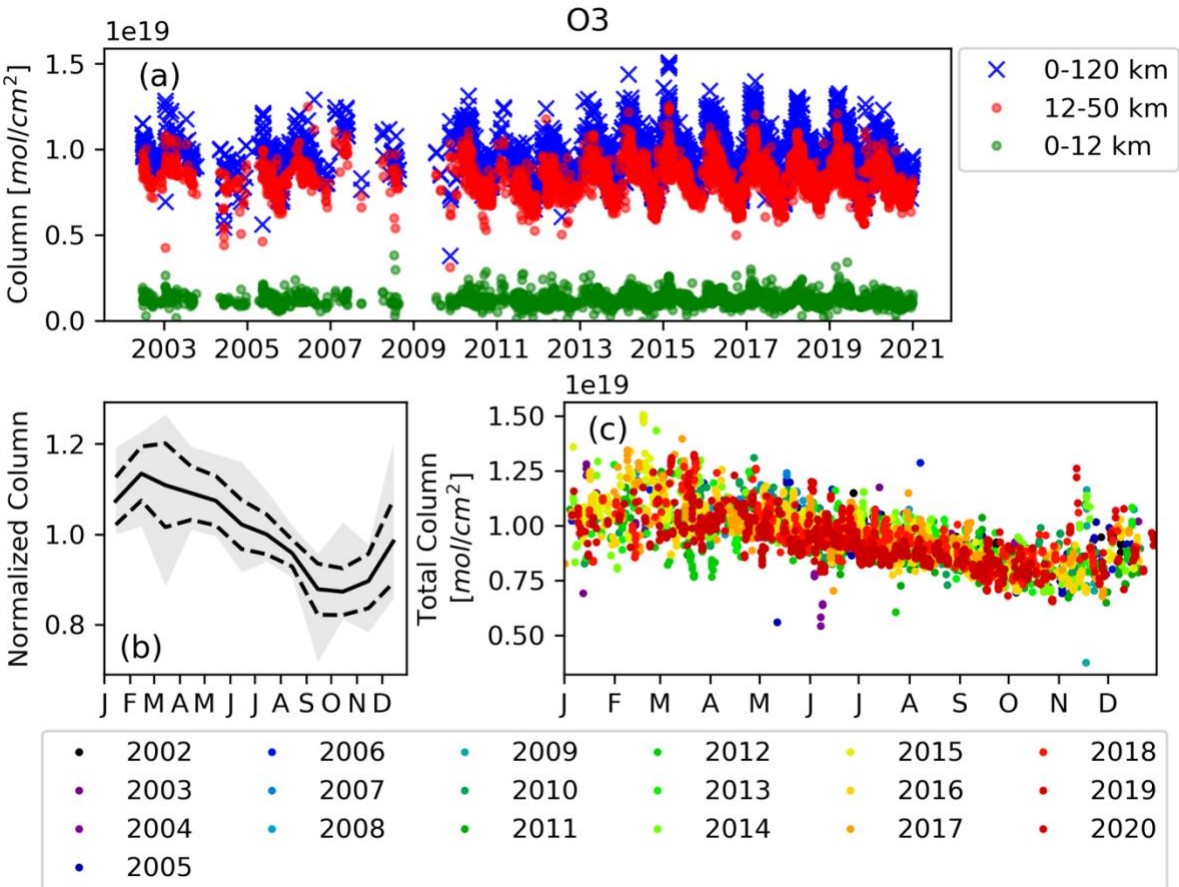

**Figure 24: O₃ measurements at TAO from 2002-2020. (a) The total column (blue) and stratospheric (red) and tropospheric (green) partial column time series. (b) Mean relative monthly means, calculated by normalising the monthly means with the respective annual mean over the 2002-2020 dataset, and averaging (e.g., January 2003 data is normalized by the annual mean from 2003, and so on, and all January mean values are then averaged). The range (min – max) is shaded in grey and ± 1σ is indicated by the dashed line. (c) The annual cycle, showing all total column measurements, colour-coded by year.**



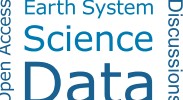

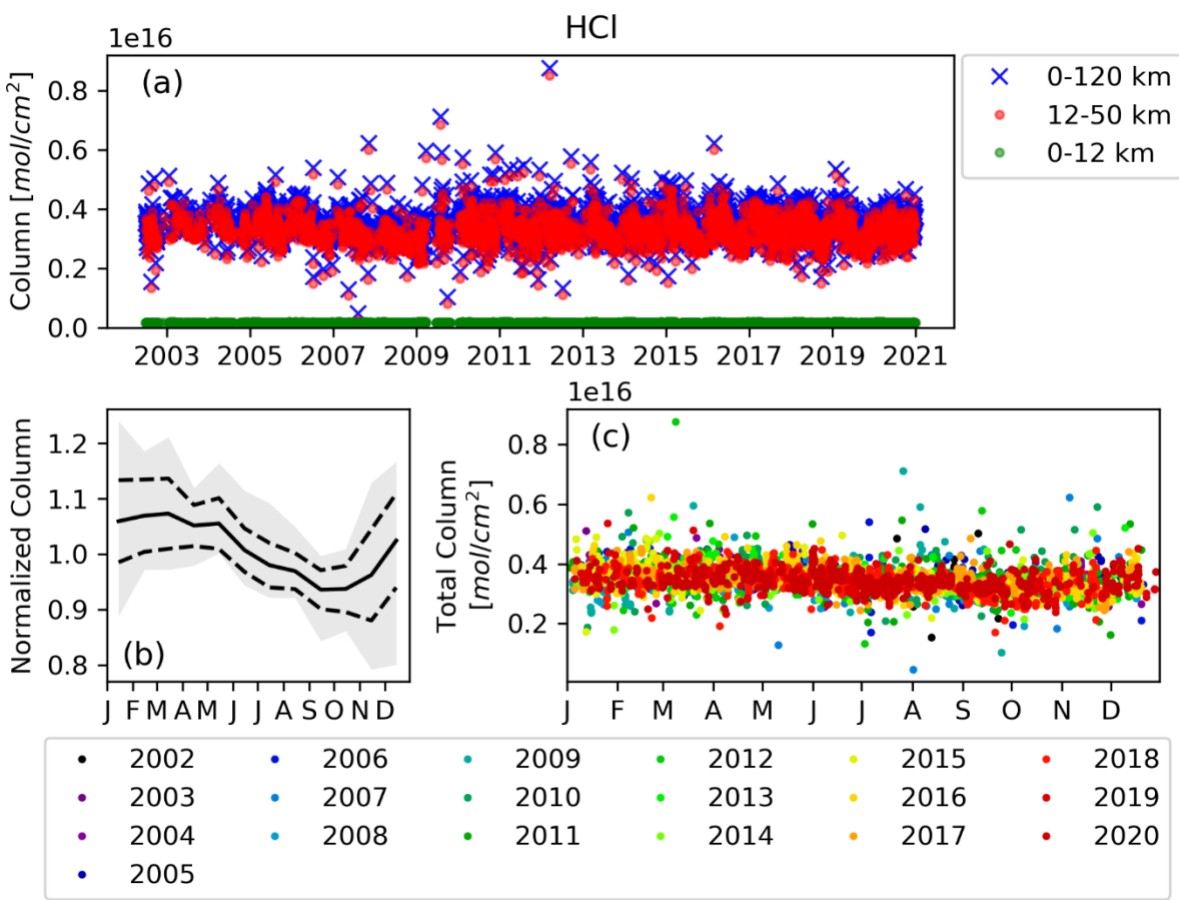

**Figure 25: Same as Figure 24 but for HCl.**

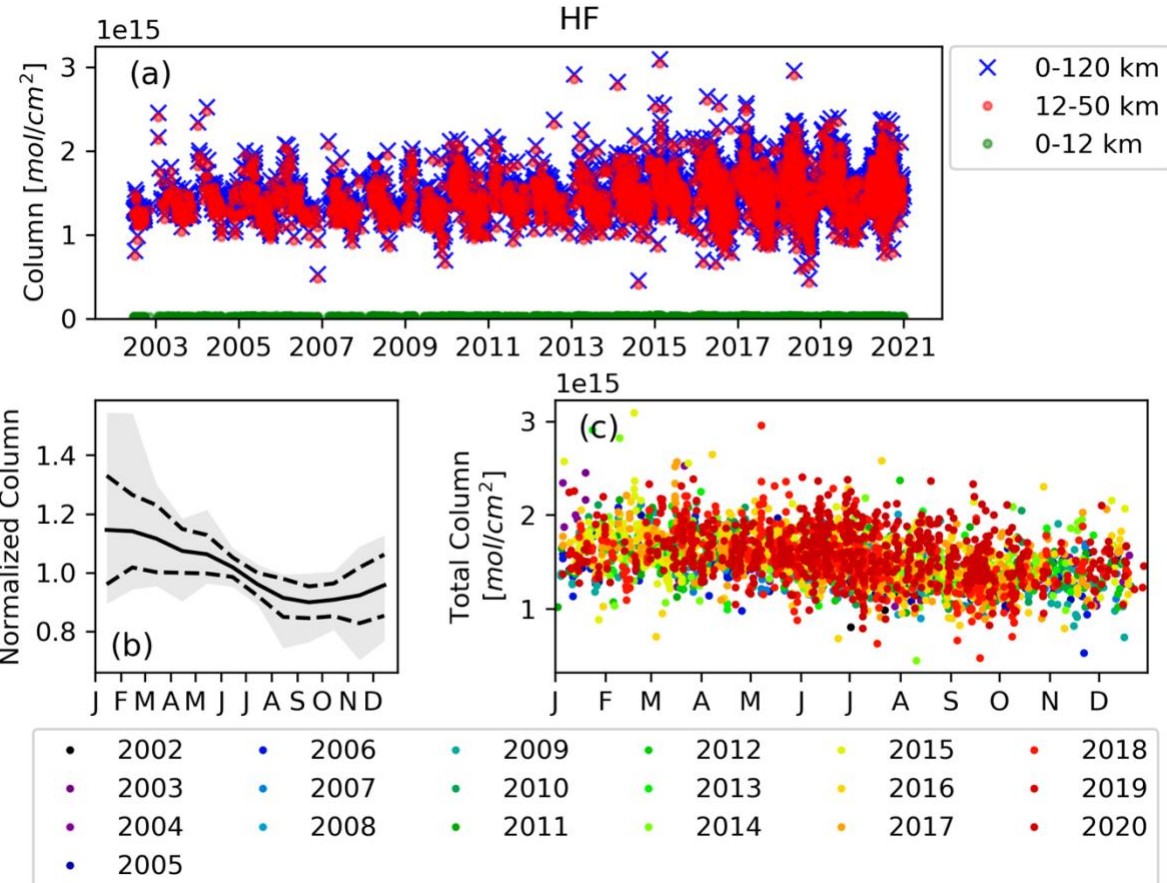

Figure 26: Same as Figure 24 but for HF.





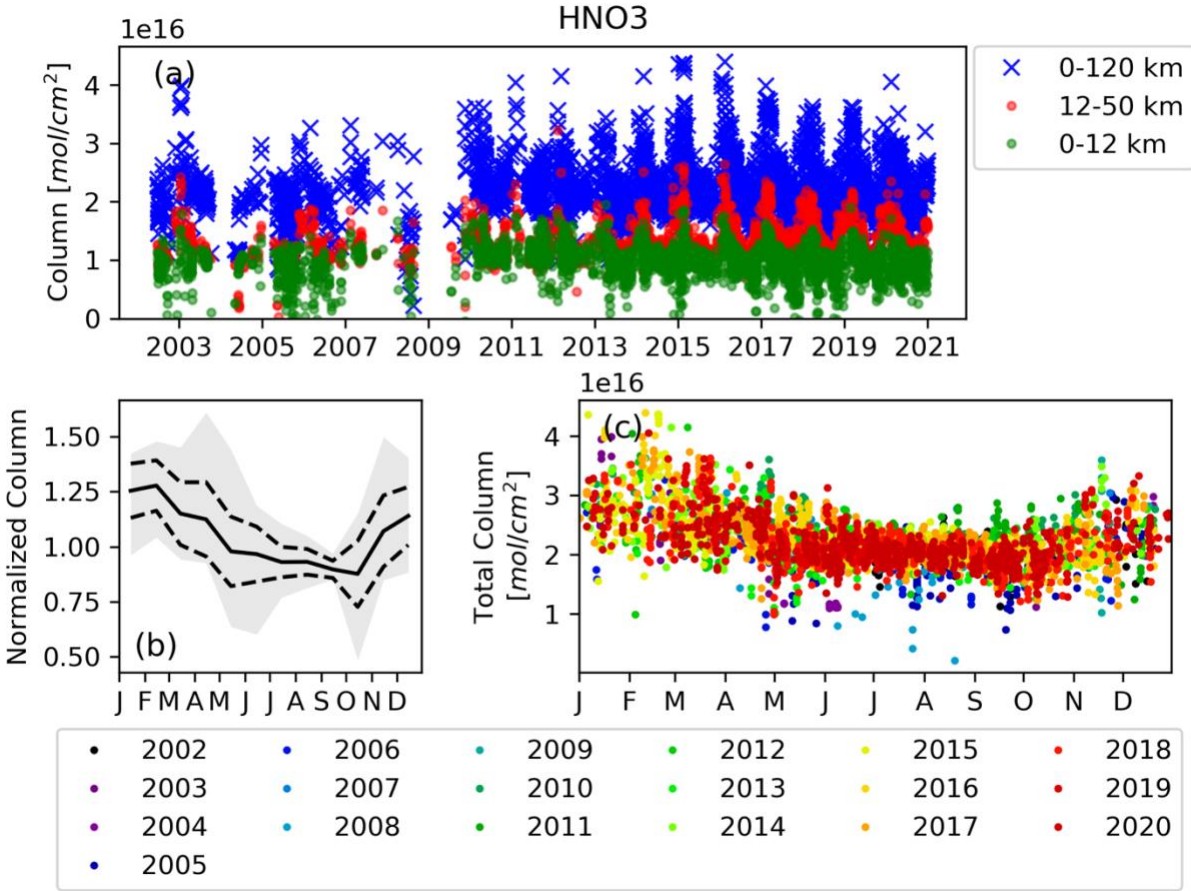

**Figure 27: Same as Figure 24 but for HNO₃.**

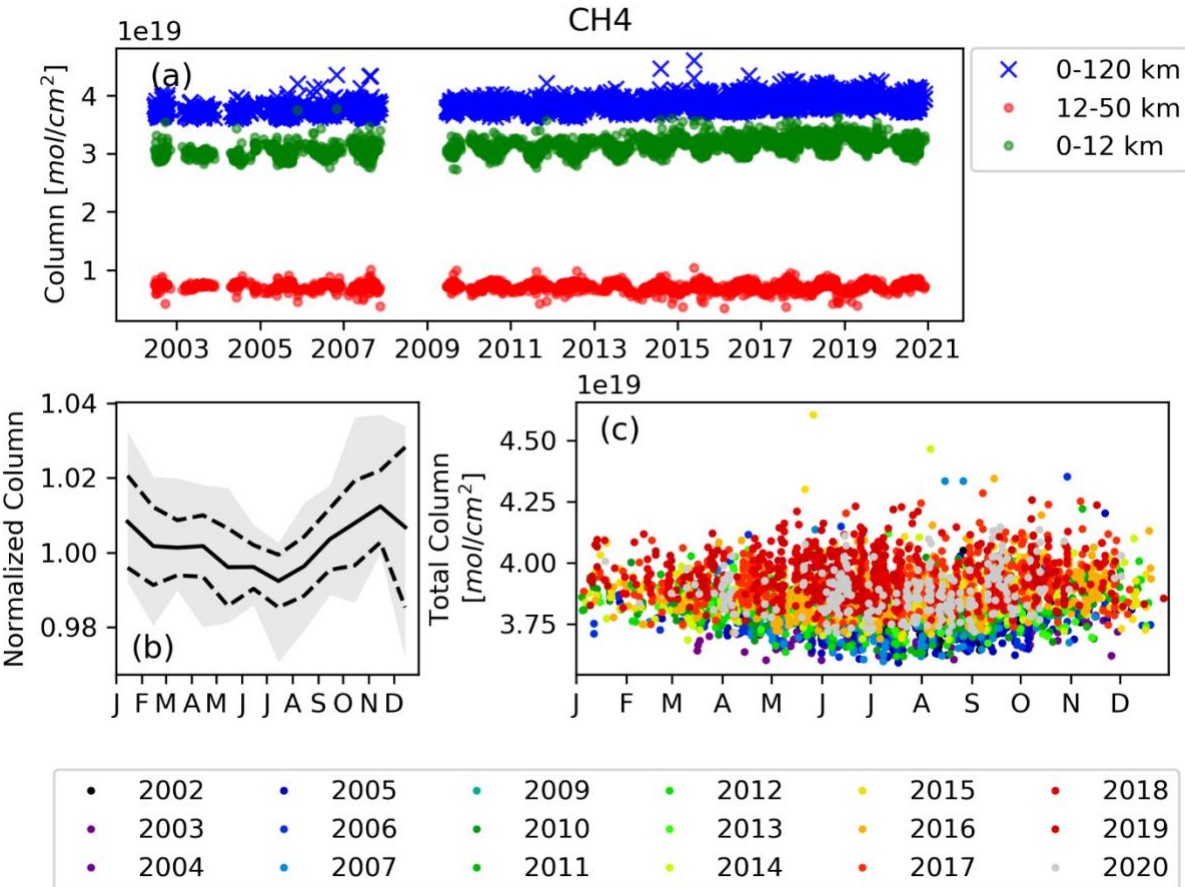

**Figure 28: Same as Figure 24 but for CH₄.**



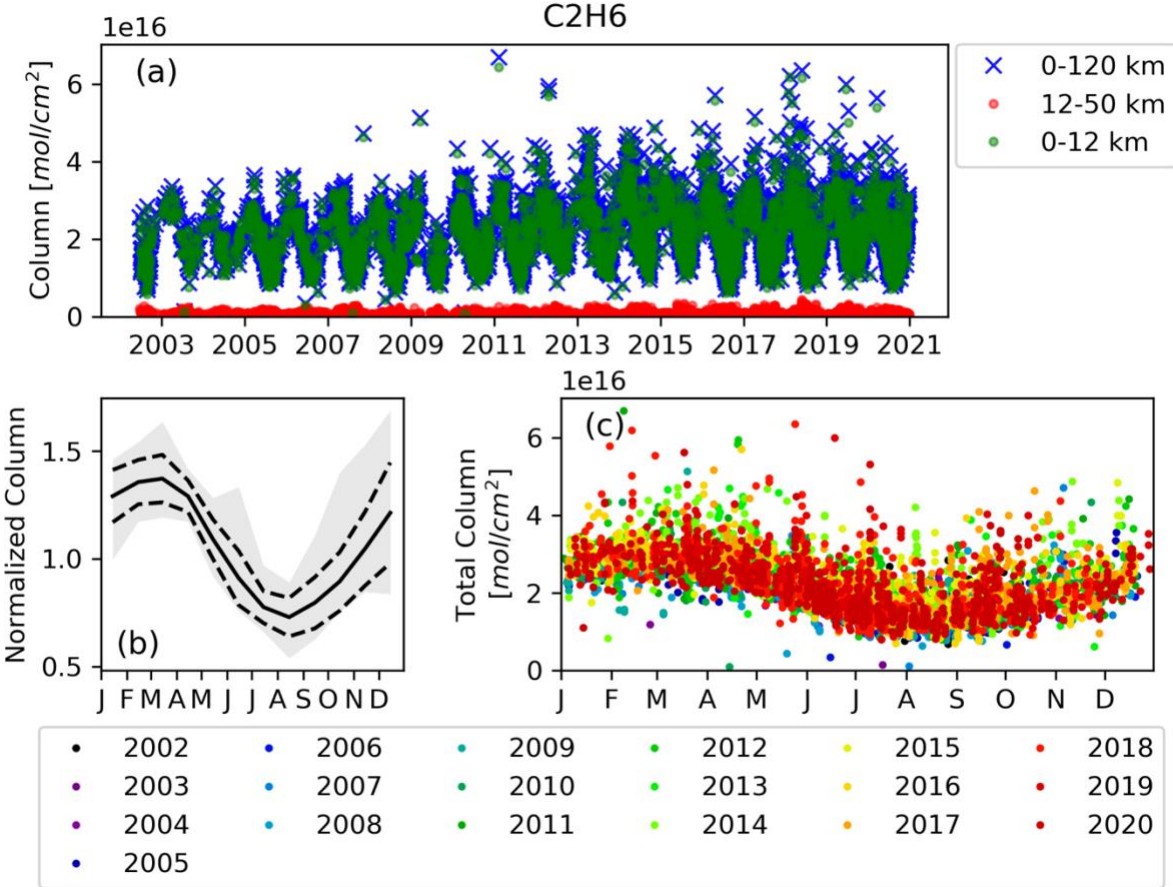

Figure 29: Same as Figure 24 but for $C_2H_6$.

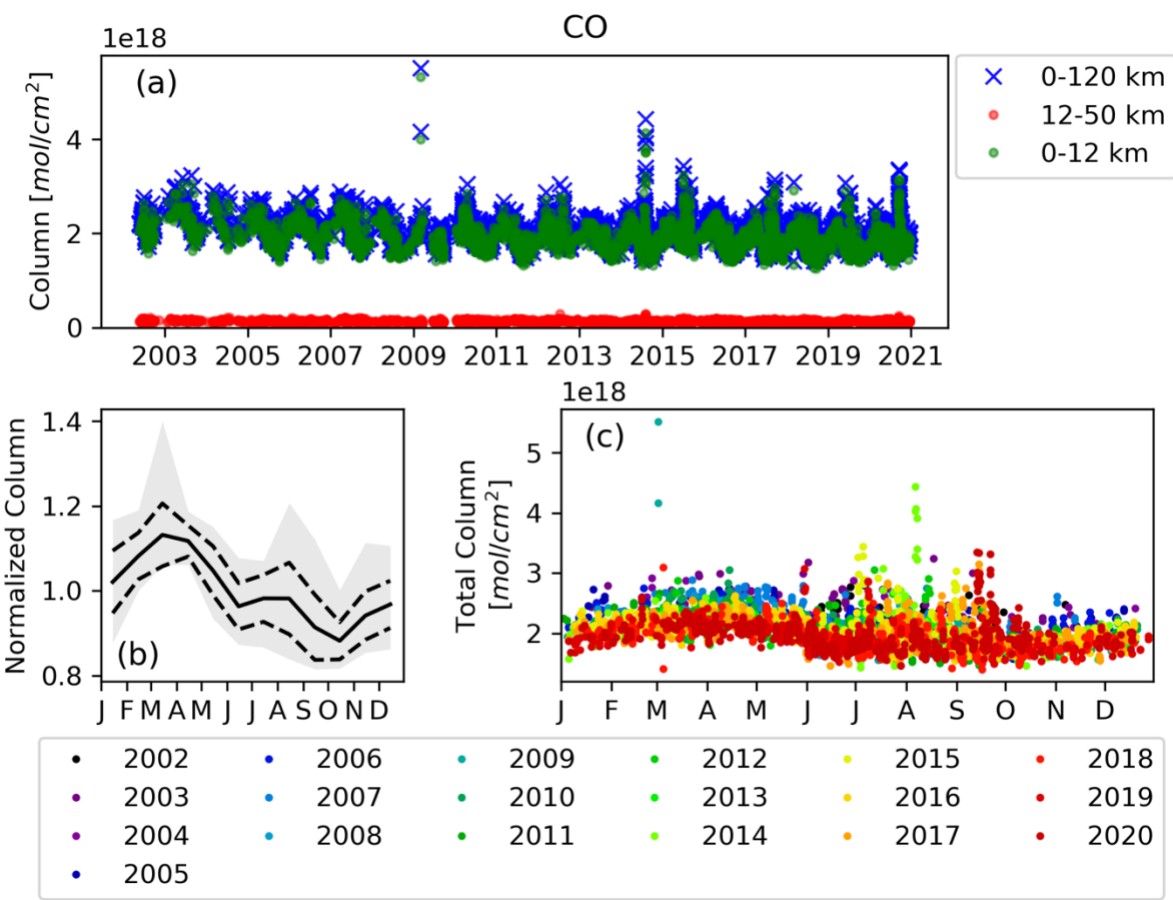

**Figure 30: Same as Figure 24 but for CO.**

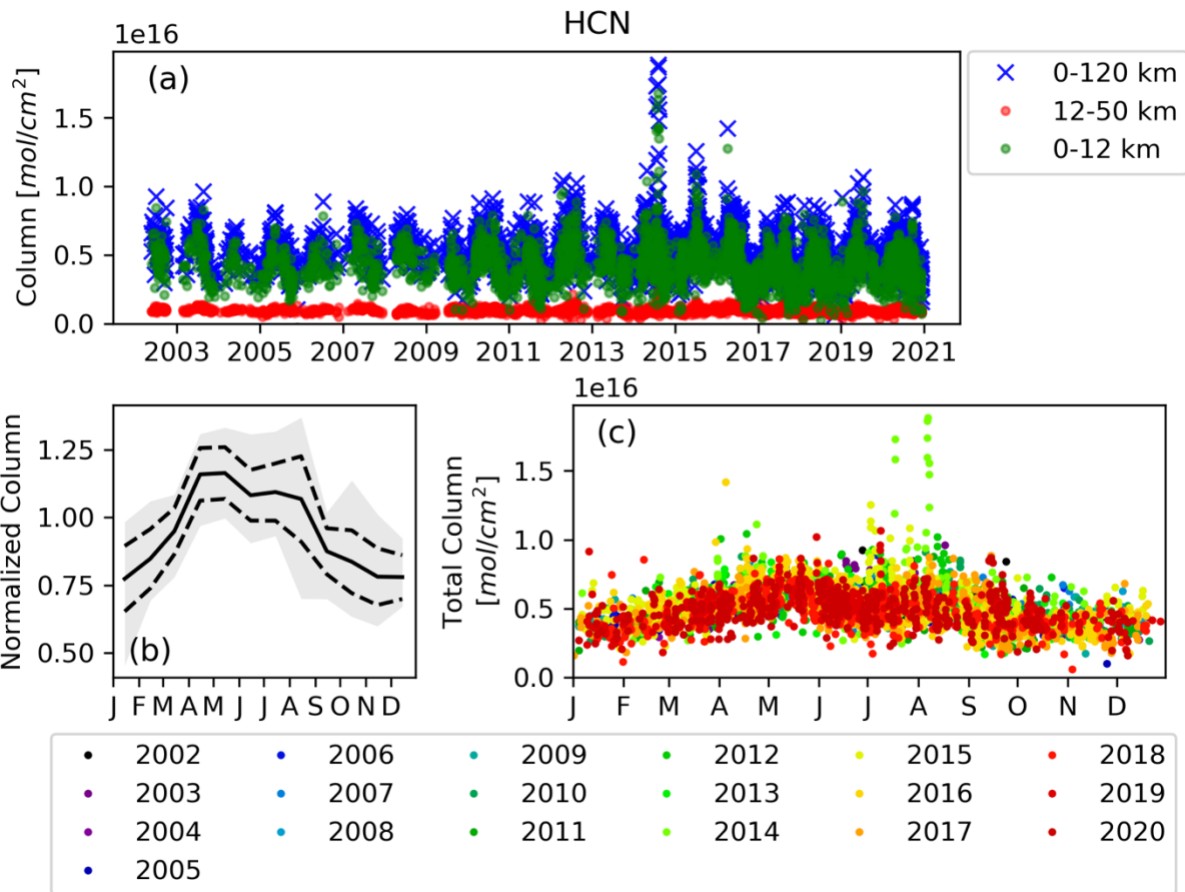

**Figure 31: Same as Figure 24 but for HCN.**





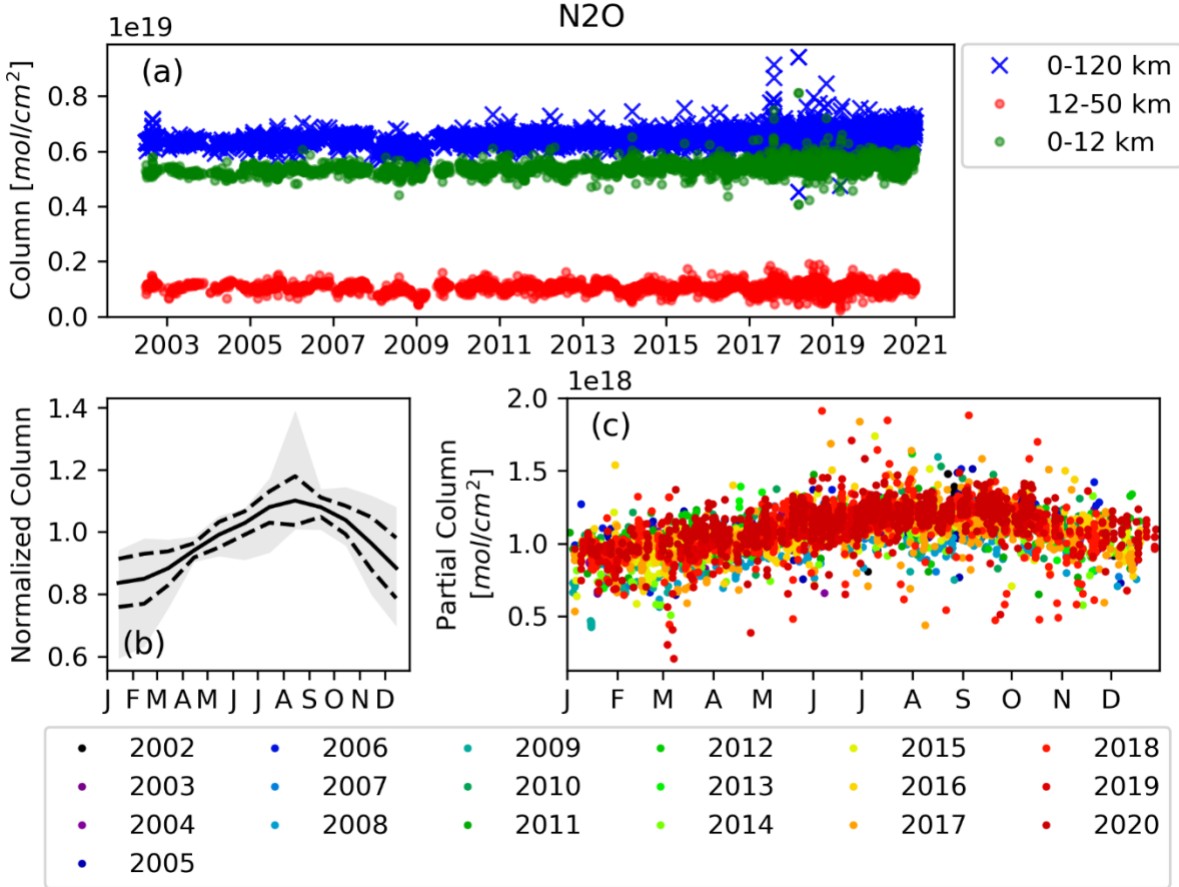

**Figure 32: Same as Figure 24 but for N₂O, except for (b) and (c) where the monthly stratospheric partial columns are plotted and the range of mean monthly values are shaded in grey to illustrate the seasonal cycle in the stratospheric partial column.**



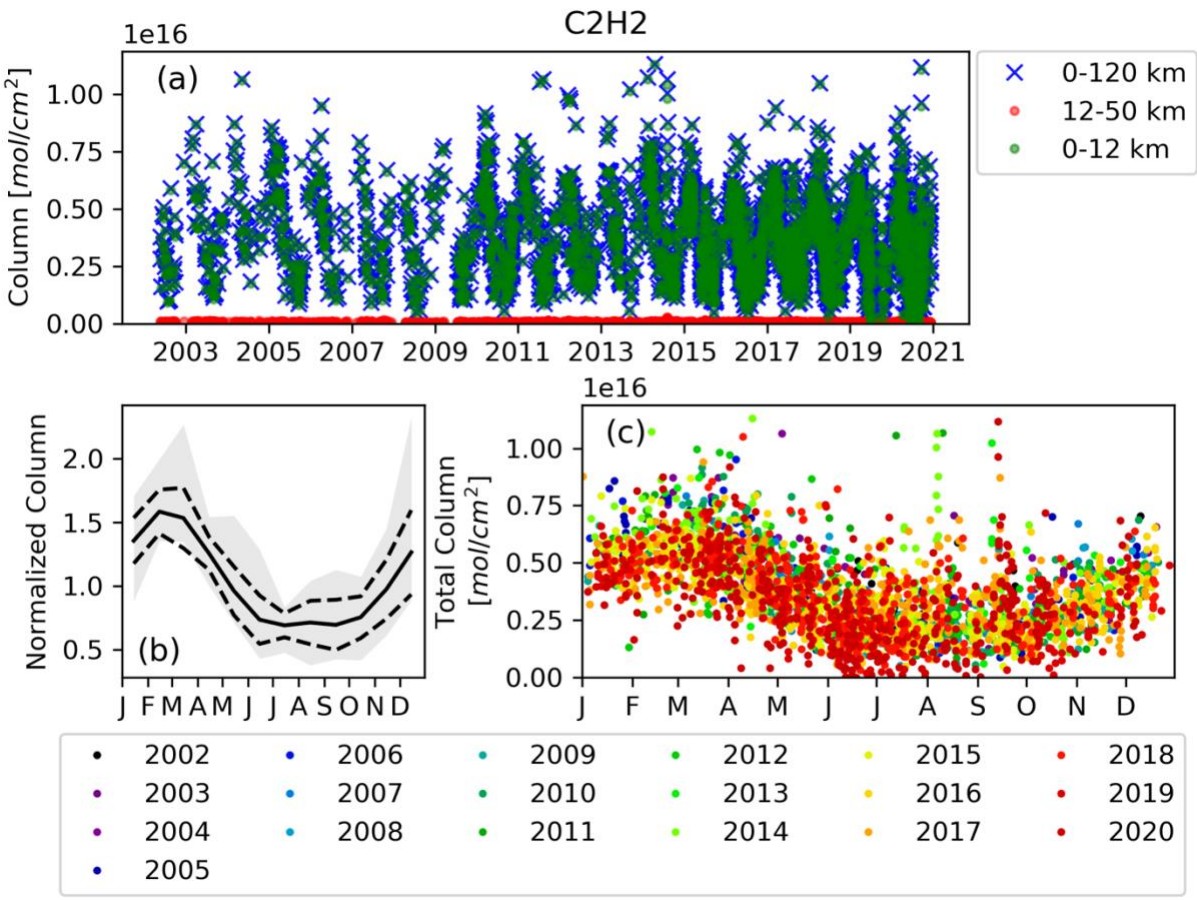

**Figure 33: Same as Figure 24 but for C₂H₂.**



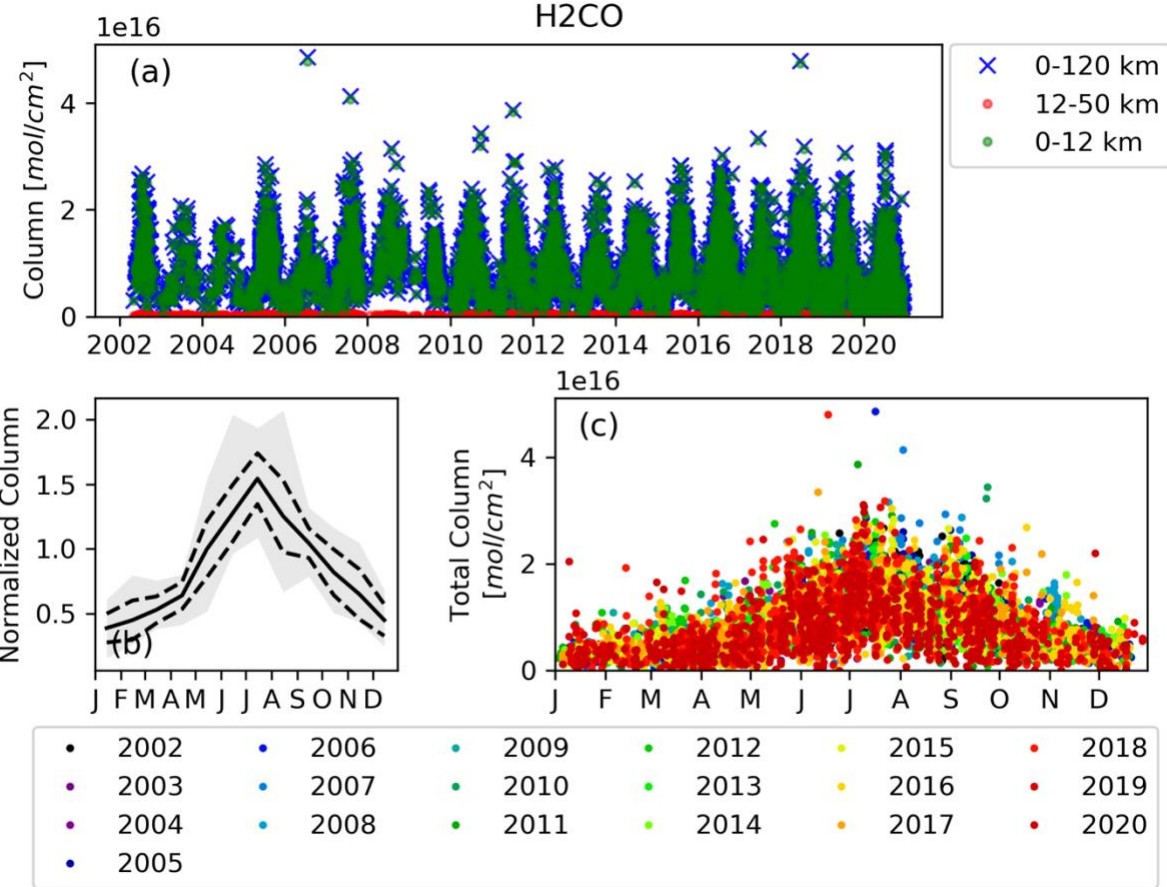

**Figure 34: Same as Figure 24 but for H₂CO.**



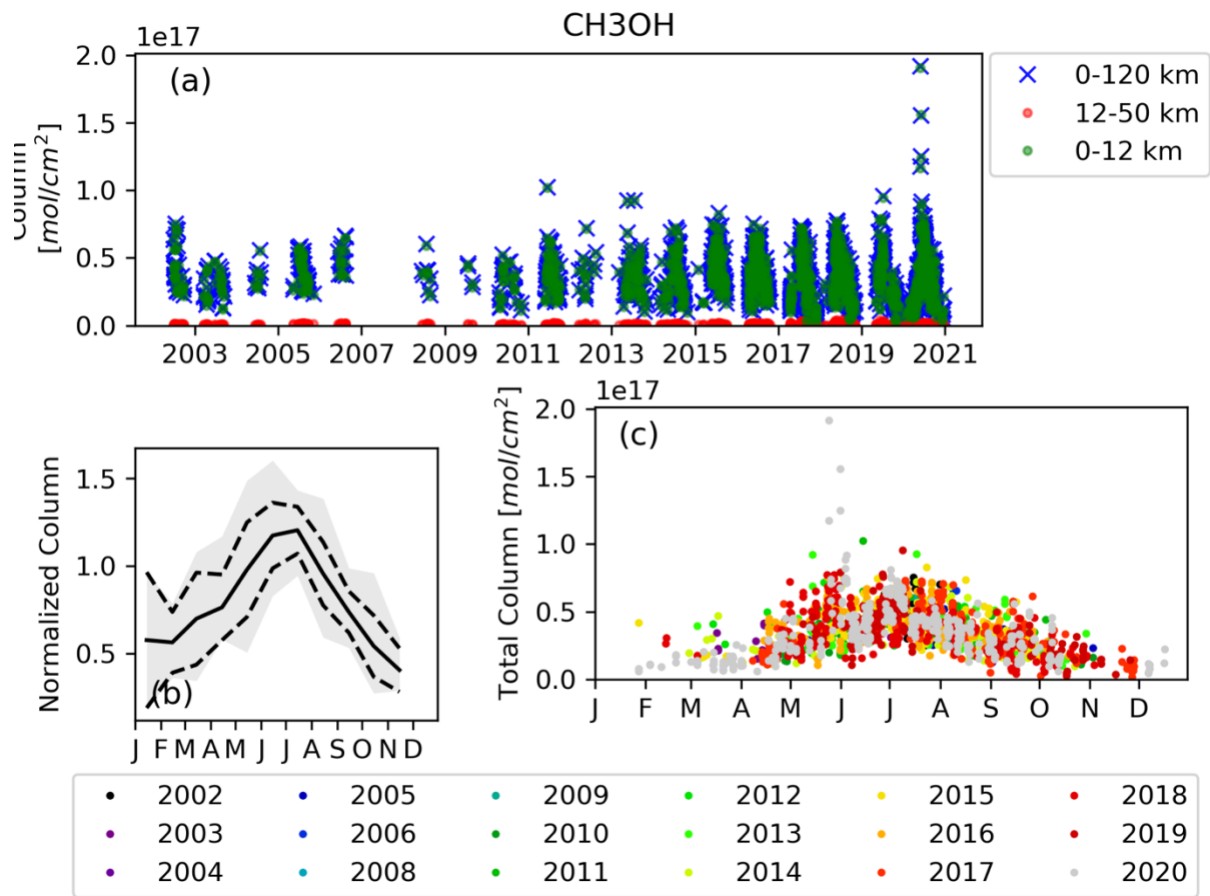

**Figure 35: Same as Figure 24 but for CH₃OH.**



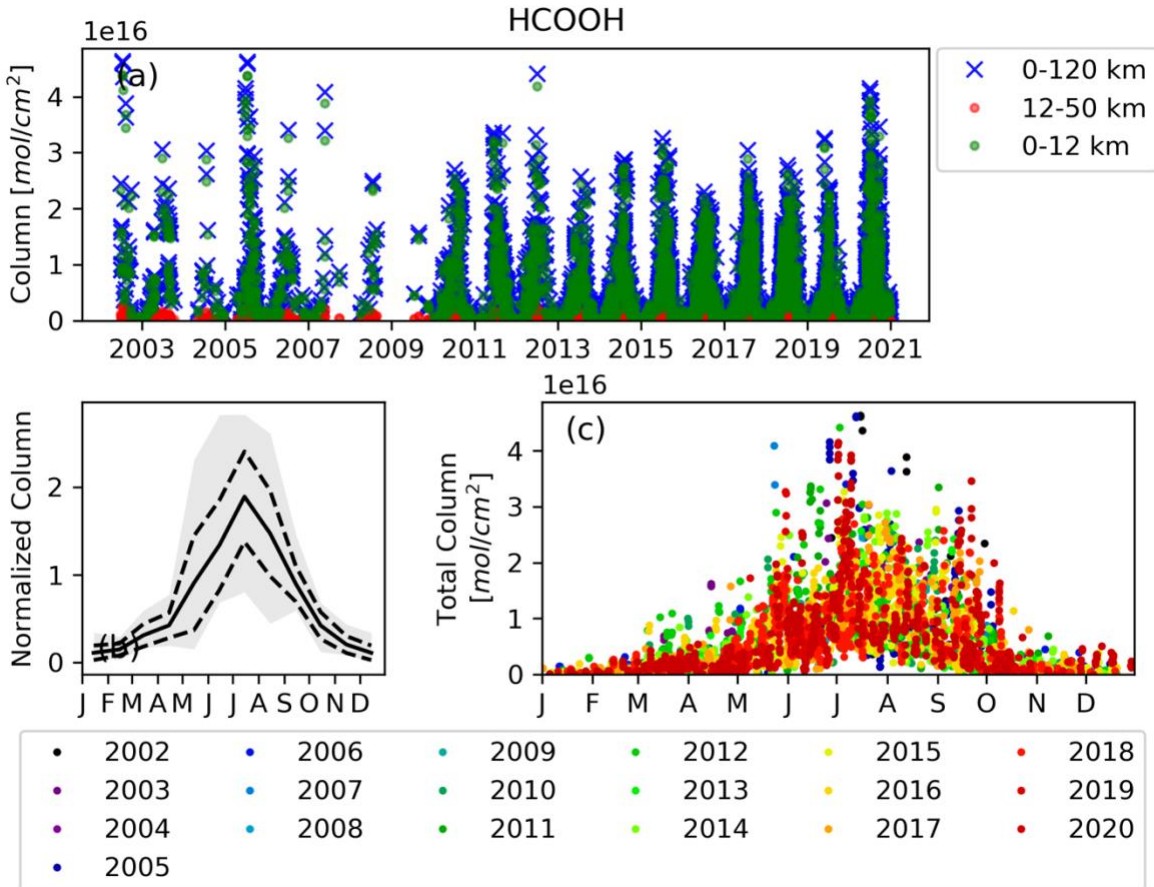

Figure 36: Same as Figure 24 but for HCOOH.

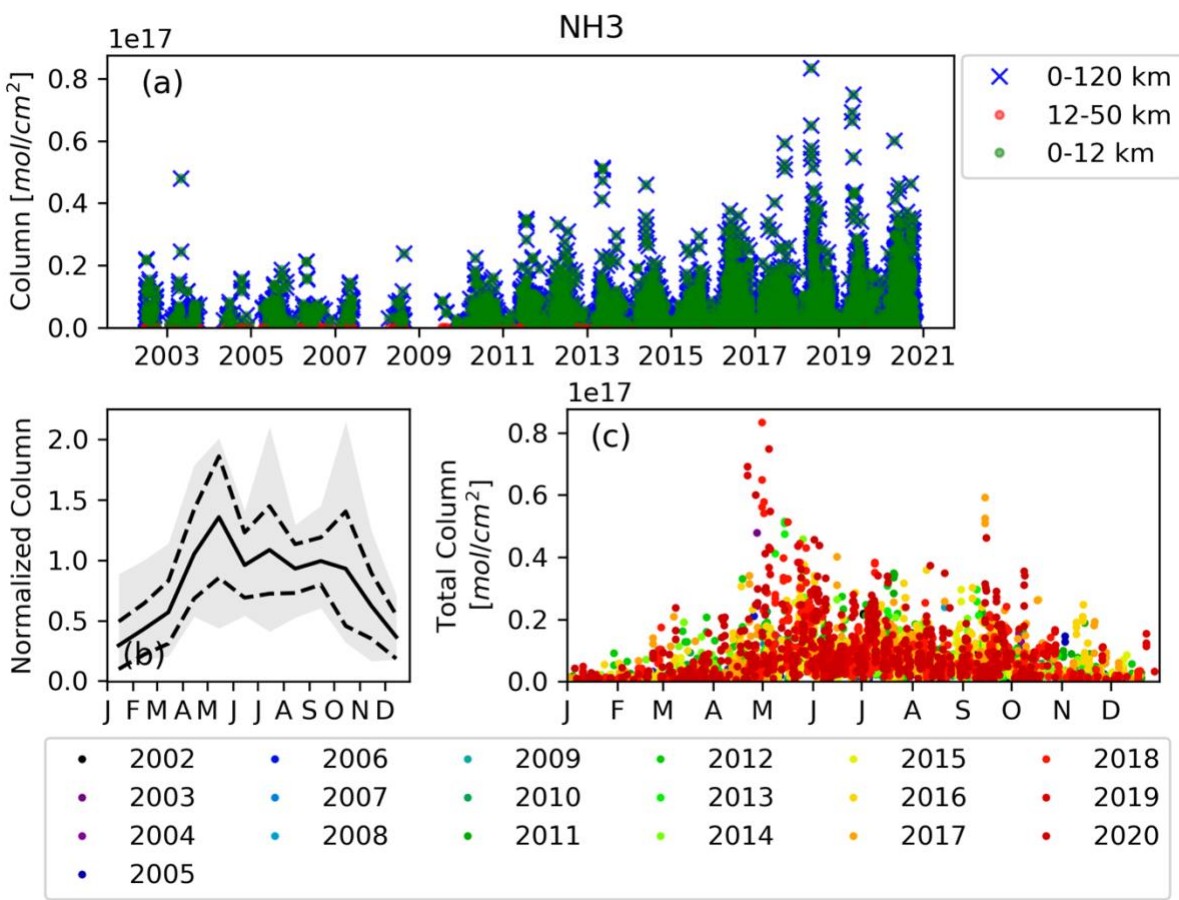

Figure 37: Same as Figure 24 but for NH₃.