# Peer review of "NDACC FTIR trace gas measurements at the University of Toronto Atmospheric Observatory from 2002 to 2020"

_Earth System Science Data, 2022_

## Author Comment (AC1)

**RESPONSE TO REVIEW 1 of Yamanouchi et al. (2023)**

We thank the reviewer for their comments on the manuscript, which we have addressed below. The comments are in blue font and the responses are in black font.

This paper is by a well respected group, with high technical capability in ground based remote sensing. The manuscript is well written, clearly states the methodology, measurement outcomes, and in general a high standard of presentation. This is an important dataset for Canada, and while the dataset itself could be reproduced in principle, it covers a long timeline, and the level of technical exertise means that it is very unlikely to repeated in the Canadian region.

The data has already been used a number of studies, satellite validation, and will continue to be an important part of future model and other measurement comparisons.

While the FTS instrumentation is a Canadian domestic system (Bomem), and is not used widely in other equivalent NDACC sites on a global basis, the Toronto group is very experienced in maintaining and operating this FTS at an acceptable level within the network. In particular, the use of HBr internal cells to monitor the instrument lineshape is an important determination of the stability of the instrument. This has been done since the beginning of the measurement record. It is clear from this record that the FTS has been upgraded at least once, 2014 for example, so this may have had an impact on the data record. The authors note that there was no impact on the retrieved columns; is this a qualitative or quantitative statement?

The analysis method is state of the art, in terms of the software package SFIT4, and each species that is retreived follows the agreed protocol that has been painstakenly developed by members of the NDACC community. The Toronto group regularly reports on all standard NDACC (via archiving) as well as a number of other interesting research gases.

The presentation quality and written components are in general very good. Below is a short list of minor corrections for consideration.

line 105: is the data for the low modulation periods included in the data record? Are they flagged in any way?

Data with low modulation efficiency are included in the data record and are not flagged in the NDACC archive. After the 2014 upgrade to the FTIR, we saw no qualitative impact on retrieved columns.

line 146: missing bracket

Fixed.

line 158:  "...2016. These..."

Fixed.

line 165: O2?

Fixed.

line 308: wayward full stop

Fixed.

line 358: there is also natural background HCl from the ocean release of CH3Cl

This information has been added.

table 2: have the authors considered using the second HNO3 window simultaneously (872-874 cm-1) with the 868cm-1 region?

No, the $HNO_3$ retrievals have only been performed using the microwindow at 867.50-870.00 $cm^{-1}$ as recommended by the NDACC IRWG. The 872-874$cm^{-1}$ could however be tested in the future.

figure 1: odd mixture of metric and imperial units?

This is due to the figure being adapted from the manufacturer's schematics.

figure 18: wavenumber scale missing

Fixed.

Figure 28: the key colour for 2020 (both CH4 and CH3OH) is grey but for all other graphs it is red?

This is because we do not have data for those species in 2008.

---

## Author Comment (AC2)

**RESPONSE TO REVIEW 2 of Yamanouchi et al. (2023)**

We thank the reviewer for their comments on the manuscript, which we have addressed below. The comments are in blue font and the responses are in black font.

This work by Yamanouchi et al. entitled "NDACC FTIR trace gas measurements at the University of Toronto Atmospheric Observatory from 2002 to 2020" presents data of a large set atmospheric trace gases derived from mid-infrared solar absorption measurements collected in Toronto during nearly two decades in the framework of the Network for Detection of Atmospheric Composition Change. Globally, only about two dozens of NDACC FTIR sites are in operation, and only a subset of these sites already have collected continuous measurements spanning such a long period. For several very minor trace gases presented by Yamanouchi et al., infrared remote sensing as performed by the NDACC FTIR sites is the primary tool for atmospheric composition monitoring, so the data set presented here is highly relevant.

The description of the data set is quite accurate, however, I have a couple of suggestions for technical improvements, which should be taken into account in the final version of the manuscript. My detailed comments are provided below.

Abstract: "the retrievals have been optimized" – does this imply that the applied retrieval recipes differ from the procedures as currently recommended by NDACC? - Please clarify.

Optimized here means that the spectrometer was run with the mentioned species in mind, using filters that are best suited for retrieving absorption lines of the species, and following NDACC IRWG recommendations.   However, we have changed "optimized" to "performed" in case the former implies a rigorous optimization procedure.

Section 2.1: It is mentioned that the transition to higher scan speed allows to collect more measurements per day. This implies that the number of coadded scans has been kept constant, not the total integration time. Would you please specify integration time per spectrum before and after the change? (Reading further, I found this issue is discussed further down, line 285ff, but it would better fit in this section.)

Integration time is now mentioned in Section 2.1, as well as a statement that there are further details in Section 3.2.

Section 2.1: the description is not clear about whether interferograms from both detectors can be collected at the same time, or whether these need to be recorded sequentially.

It is implied from the statement about measuring sequentially through filters; Clarification has been added.

Section 2.1: Are AC or DC coupled interferograms recorded?

The system is not set up to record DC-coupled interferograms.

Section 2.1: In addition to table 1 summarizing the basic filter characteristics, it would be nice to add a figure collecting low-resolution blackbody or globar spectra for each filter (as there are significant variations between different filter batches and the resulting spectral envelope also depends on detector and beamsplitter characteristics).

While this would be a good datapoint to have and discuss, it is not something that has been done at our site.

Section 2.1: It is laudable that regular cell measurements are collected and have been carefully analysed for documenting the alignment status of the spectrometer. However, from the description it is not clear to me which strategy is used for taking into account the imperfect ILS indicated by the cell measurements – are the resulting time series of modulation efficiency amplitude and phase error applied in the atmospheric retrievals? If so, are the lab results smoothed / averaged and the resulting values used for certain sub-periods? Which assumptions have been made concerning extrapolation of the measured ILS characteristics as function of wavenumber (one might assume that ILS disturbances increase with wavenumber)?

The ILS measurements are used to assess the health of the spectrometer and determine what maintenance (e.g., re-alignment) is needed. The ILS derived from the cell measurements is not routinely included in the atmospheric retrievals given issues with variability and extrapolation to other wavenumbers and as no significant differences in column amounts were found when testing three implementations of the ILS in the spectral fitting (Wunch et al., 2007; Taylor, 2008).

Section 3.1: I assume that the SNR for a given optical filter as used by the retrieval is allowed to vary between individual spectra, and this variable SNR (evaluated in a suitable spectral range for each filter) is used in the retrieval, correct?

Yes, this is correct. The manuscript states:

"The signal-to-noise ratio of the measurements (SNR) is the ratio of the maximum signal in the selected microwindow to the noise level for each spectrum (the standard deviation of the zero-level signal in a microwindow) and provides a measure of the quality of the spectrum."

Line 237 "The averaging kernels, the rows of the averaging kernel matrix …" -> "the rows of the averaging kernel matrix …"

Fixed.

Line 267ff: I think we should reserve the term "null space error" for the errors introduced by the discretization of the altitude coordinate, while the "smoothing error" is dominated by the limited vertical resolution of the measurement (assuming a sufficiently fine vertical

discretisation). If the characteristics of atmospheric variability (a-priori covariances) are derived in a self-consistent manner from a given reference ensemble of profiles, which are sampled on a very fine vertical grid, then the smoothing error (of e.g. a partial column confined by two selected altitude values) will be independent of the chosen vertical discretization.

We have removed the reference to null-space error.

Line 285ff: The construction of the statistical error budget seems incorrect to me. Merging the erratic pointing offsets of the tracker with the SZA smear due to integration time results in a significant overestimation of this source of statistical error. The SZA smear generates a systematic error contribution (as it will generate the same bias for an imaginative repeated measurement in the same dSZA / dt range). Ok, one might claim that during a day various dSZA / dt situations are sampled and when doing the statistics over this set of measurements in a day, this will resemble a statistical error – but still: why should we ascribe this large error on measurements taken around noon?

While we agree that the SZA uncertainty can include both random and systematic components, some NDACC FTIR stations assign it to both (e.g., Zhou et al., 2021, https://doi.org/10.5194/amt-14-6233-2021) and others just to random error (Mahieu et al., 2021, https://doi.org/10.1525/elementa.2021.00027). Given the variation in sampling SZA over the course of a day and a year, we have treated this as a random error. We have chosen a conservative approach to quantify the maximum uncertainty due to the SZA varying during a measurement, basing it on the average change in SZA over the integration time throughout the year.

Line 291ff: "Three values describe the line parameters in the forward model, each with its own uncertainty: the line intensity, the temperature-broadened half-width, and the pressure-broadened half-width." Please reword, there is no "temperature-broadened half-width". Also be aware that there are two kinds of errors here (related to temperature): the modelled line width might be incorrect because the assumed temperature is incorrect or because the assumed temperature dependence of the broadening is incorrect.

The temperature-broadened half-width was reworded to temperature dependence error to be more consistent with HITRAN parameters. Reference to temperature errors have been fixed throughout the manuscript.

Line 308: remove "."

Fixed.

Concerning the gas specific systematic column errors and the associated description in the per-species paragraphs: please recheck the descriptions carefully, they cannot be correct! Example (1) HCOOH: "For total columns, the mean systematic error is 10.41 % ... the systematic error is dominated by the pressure-broadening error". However, table 3 states that the assumed line

intensity uncertainty is 7.5%. As a line intensity error propagates directly into the column error (note: DOFs ~1) and the total systematic error is 10.41%, this needs to be the leading systematic error source. Example (2) N2O: "For the total columns, the mean systematic error is 3.59 % and is dominated by the temperature error" – this cannot be true, as the assumed line intensity error is 3.5% (table 3).

Fixed (reference to pressure broadening error removed from HCOOH, N2O systematic error now mentions line intensity error).

For very weak absorbers (e.g., H2CO and HCOOH) with large atmospheric variability, the random error budget should be expressed as a column value (the detection threshold), not as a relative error. Given the considerable noise error contribution for these species: why do no negative columns occur in the time series (or are they simply clipped in the figures?)?

The time series plots do have zero as the lower limit, however, many of the negative columns are removed during the RMS/DOFS ratio filtering process described in Section 3.3 (as well as the in filtering done to meet the CAMS standard described in the same section). While we agree that detection threshold is a good way to express errors for these species, we went with the relative error to be consistent with the other retrieved species.

In my pdf document, figure 7 starts with the HCN panel, according to the figure caption, there should be additional gases?

Figure 7 is complete in the submitted manuscript, see at https://essd.copernicus.org/preprints/essd-2022-470/essd-2022-470.pdf.

The spectral fit plots need to be revised, often the residual ordinate is not readable, sometimes meaningless (just a single zero mark), and the spectral abscissa values often (nearly) overlap.

The plots were fixed by using more appropriate scales.

Figure 22: The C2H6 total column sensitivity looks strange (sharp notch around 20 km). Perhaps for some species cease the lines before the altitude range is entered where the calculation by the analysis code becomes unreliable. See, e.g., the C2H2 sensitivity (sudden drop to zero at 20 km).

While we agree that some of the sensitivity plots do appear strange, we believe this figure is appropriate, as we archive all the column data (up to 120km), including the partial columns and averaging kernels, regardless of the sensitivit. Since the figure (and indeed the paper) is to illustrate the dataset, we believe this is appropriate given the context.